# Residence times of air in a mature forest: observational evidence from a free-air CO₂ enrichment experiment

Edward J. Bannister[1,3,4], Mike Jesson[2,1], Nicholas J. Harper[1], Kris M. Hart[1], Giulio Curioni[1], Xiaoming Cai[5,3], A. Rob MacKenzie[1,3]

[1] Birmingham Institute of Forest Research, University of Birmingham, Edgbaston, Birmingham, United Kingdom
[2] Department of Civil Engineering, University of Birmingham, Edgbaston, Birmingham, United Kingdom
[3] Department of Geography, Earth, and Environmental Sciences, University of Birmingham, Edgbaston, Birmingham, United Kingdom
[4] Now at Risk Management Solutions, London, United Kingdom
[5] Retired

*Correspondence to*: Rob MacKenzie (a.r.mackenzie@bham.ac.uk)

**Abstract.** In forests, the residence time of air—the inverse of first-order exchange rates—influences in-canopy chemistry and the exchanges of momentum, energy, and mass with the surrounding atmosphere. Accurate estimates are needed for chemical investigations of reactive trace species, such as volatile organic compounds, some of whose chemical lifetimes are in the order of average residence times. However, very few observational residence-time estimates have been reported. Little is known about even the basic statistics of real-world residence times or how they are influenced by meteorological variables such as turbulence or atmospheric stability. Here, we report opportunistic investigations of residence time of air in a free-air carbon dioxide enrichment (FACE) facility in a mature, broadleaf deciduous forest with canopy height $h_c \approx 25$ m. Using nearly 50 million FACE observations, we find that median daytime residence times in the tree crowns range from around 70 s when the trees are in leaf to just over 34 s when they are not. Residence times increase with increasing atmospheric stability, as does the spread around their central value. Residence times scale approximately with the reciprocal of the friction velocity, $u_*$. During some calm evenings in the growing season, we observe distinctly different behaviour: pooled air being sporadically and unpredictably vented—evidenced by sustained increases in CO₂ concentration—when intermittent turbulence penetrates the canopy. In these conditions, the concept of a residence time is less clearly defined. Parameterisations available in the literature underestimate turbulent exchange in the upper half of forest crowns and overestimate the frequency of long residence times. Robust parametrisations of residence times (or, equivalently, fractions of emissions escaping the canopy) may be generated from inverse gamma distributions, with the parameters $1.4 \leq \alpha \leq 1.8$ and $\beta = h_c/u_*$ estimated from widely measured flow variables. In this case, the mean value for $\tau$ becomes formally defined as $\bar{\tau} = \beta/(\alpha - 1)$. For species released in the canopy during the daytime, chemical transformations are unlikely unless the reaction time scale is in the order of a few minutes or less.

## 1 Introduction

Forests cover nearly a third of the Earth's land surface and exchange momentum, energy, and mass with the atmosphere. Forest-atmosphere exchanges are fundamental to forest ecology, involving transfers of water vapour, carbon dioxide (CO₂), trace gases including biogenic volatile organic compounds (BVOCs), and particles such as pollen and spores. Forest-atmosphere exchanges also influence air quality, meteorology, and the climate, for example, through BVOCs interacting with oxidants such as O₃ and OH (Fuentes et al., 2000; MacKenzie et al., 2011; Peñuelas and Staudt, 2010; Pyle et al., 2011; Rap et al., 2018).

Turbulent motions transport the air from the boundary layers around the forest elements into the canopy airspace and out into the surrounding atmosphere. The properties of these turbulent motions depend on factors such as a forest's structure and the

atmospheric conditions (Bannister et al., 2022; Brunet, 2020; Finnigan, 2000). The turbulent exchange determines the extent to which a forest is ventilated, i.e., how quickly the air within the forest is replaced by air from the surroundings. The rate at which a forest is ventilated is especially pertinent when considering reactive compounds, such as many BVOCs, whose chemical lifetimes can be in the order of a few minutes (Kesselmeier and Staudt, 1999). In this context, it is helpful to consider a 'residence time', which refers to a representative amount of time air parcels spend within the forest air space. During this time, the air parcels can exchange mass with the forest and one another, and the gases within them may participate in chemical reactions. Accurate estimates of residence times in forests are needed to scale leaf-level chemistry and meteorology to the regional and global scales relevant to commerce and policy (Forkel et al., 2015; Guenther et al., 2012). Residence times and other time-scale estimates are commonly used in urban studies, for example, to quantify how well a city is ventilated, or the time over which pedestrians are exposed to pollutants (e.g., Cai, 2012; Lau et al., 2020; Lin et al., 2014; Lo and Ngan, 2017).

There is no single definition of a residence time for air in forests. The first attempts to investigate the statistics of air parcels in forests adopted a Lagrangian stochastic (LS) approach, by calculating statistics on a large number of air parcels within a flow (Fuentes et al., 2007; Strong et al., 2004). These LS modelling studies suggest that air-parcel residence times depend strongly on the parcel's release height. The mean residence times range from a few seconds, for parcels travelling from the forest crown, to several minutes for parcels travelling from near the forest floor (Fuentes et al., 2007; Strong et al., 2004). Long residence times—ten minutes or more—have been calculated to occur almost exclusively for parcels travelling from the lower third of the canopy.

Gerken et al. (2017) (hereafter GCF17) offer the most complete statistical account of air-parcel residence times in forests under neutral atmospheric conditions. GCF17 propose an elegant model for the distribution of residence times by adapting the inverse-Gaussian distribution and representing turbulent transport using eddy-diffusivity closure. Katul et al. (2005) used a similar approach to model the long-distance dispersion of light seeds, again under neutral atmospheric conditions. The residence times, $\tau$, have a probability density function (PDF) given by the distribution of first passage through a plane at $z = h_c$, where $h_c$ is the mean height of the forest. For a given release height, $z_{rel}$, the PDF is

$$p(\tau; z_{rel}) = \frac{|h_c - z_{rel}|}{\sqrt{4\pi K_{eq}}} \tau^{-\frac{3}{2}} \exp\left[-\frac{(h_c - z_{rel})^2}{4K_{eq}\tau}\right],\tag{1}$$

where $K_{eq}$ is a constant eddy diffusivity at each $z_{rel}$ (but may differ for different $z_{rel}$). GCF17 use Eq. (1) to define turbulent transport time scale

$$\tau_{turb}(z_{rel}) = \frac{(h_c - z_{rel})^2}{4K_{eq}(z_{rel})}.\tag{2}$$

Equation (1) predicts an exponential increase in probability with increasing $\tau$, followed by a heavy-tailed $\tau^{-3/2}$ power-law decrease beyond the mode (i.e., as $\tau$ becomes large relative to $\tau_{turb}$, the exponential term approaches unity). In forests and other plant canopies, eddy-diffusivity closure is imperfect and may be unsuitable for certain applications (Bannister et al., 2022; Finnigan, 2000; Monteith and Unsworth, 2008). However, it remains widely adopted in larger scale models because it allows in-canopy turbulent transfer to be estimated from a modest number of variables, without the prohibitive computational expense of more sophisticated closure schemes. GCF17 acknowledge the limitations of eddy-diffusivity closure and find support for Eq. (1) in that it agreed quite well with results obtained using large-eddy simulations (LES) of idealised forest canopies, particularly for parcels travelling from low down in the canopy (LES does not rely on the same closure assumptions as Eq. (1)).

GCF17 find that the median values of $\tau$ range from a few seconds in the upper crowns to around 30 minutes near the forest floor, with the variability in $\tau$ decreasing rapidly with height.

LAI and leaf-area density (LAD: the distribution of leaf area with height) are important measures of the density and morphology of canopy structure (Bréda, 2003), along with their whole-plant equivalents plant area index (PAI) and plant-area

density (PAD). Leaf and plant-area indices affect the permeability of air through the canopy (Bannister et al., 2021) and therefore the value of residence times. In their idealised forest canopy, GCF17 show that the values of $\tau$ increase with the forest's LAI, other than for parcels released high in the canopy. Bailey et al. (2014) obtained comparable results in LES investigations of exchange around short, trellis-trained crops. Bailey et al. (2014) also found residence times were longer in homogeneous canopies than heterogeneous ones. However, measurement difficulties and spatial heterogeneity make it difficult

to validate these results using field observations. For example, in real forests, leaf and plant-area indices are highly variable, reflecting species distribution, stand demographics, season (for deciduous forests), and planting density, thinning, and harvest techniques (for managed forests). Windthrow and pest or disease outbreaks can also affect LAI and LAD on scales from individual trees to large forest stands (Bannister et al., 2022; Bréda, 2003). To our knowledge, the influence of canopy density on the residence time of air remains untested in real forests.


In complex terrain, pressure perturbations around hills induce dynamical patterns in the flow, and differential heating and cooling can cause gravity-driven flows along slopes. These phenomena strongly affect forest-atmosphere exchange (Bannister et al., 2022; Finnigan et al., 2020), for example, by causing preferential venting in seemingly homogeneous patches of forest (Cook et al., 2004), or evidenced in anomalous fluxes, relative to the landscape averages (Chen et al., 2020). The flow dynamics

around forested hills are complex and are sensitive to subtle changes in temperature, slope, and canopy structure—see Finnigan et al. (2020) for detail. Their net effects on residence times in forest canopies are difficult to measure robustly, particularly using field measurements. Idealised Reynolds-averaged Navier–Stokes and LES studies of flow over forested hills show residence times of air parcels emitted in the lower part of the canopy are shorter than those moving over flat terrain (Chen et al., 2019; Ross, 2011). However, there is a large spatial variability in residence times in forests over hilly terrain. Residence

times can be long, for example, when air is trapped in the lee of a hill during weak winds (Ross, 2011).

Researchers have also used Eulerian frameworks to investigate residence times of air in forests. Edburg et al. (2012) use LES to calculate mean residence times of 8.6, 3.6, and 5.6 min for ground, canopy, and mixed sources of passive scalars released in a homogeneous forest, within the range of reported values using LS models. Wolfe et al. (2011) use a simple canopy

resistance model to estimate residence times of around 2 min for a ponderosa pine plantation. Lagrangian and Eulerian approaches to investigating residence times each has strengths and weaknesses. A Lagrangian approach offers the simplest conceptual picture. For example, one can imagine an air parcel passing over a source of a BVOC, such as a sunlit leaf, then passing through the forest air space, and eventually leaving the forest. Tracking the trajectories of lots of air parcels in this way allows one to derive residence-time statistics. However, Lagrangian approaches are only feasible using numerical models,

at least at the scales relevant to flow around a forest. They therefore inherit the limitations of numerical modelling, for example, by requiring the turbulence to be specified a priori with simplified statistics (Fuentes et al., 2007; Strong et al., 2004). Some of these difficulties are bypassed by tracking Lagrangian parcels in a flow resolved using LES (e.g., GCF17). However, LES remains computationally expensive and may underestimate the total boundary-layer turbulence (Grylls et al., 2020). LES is also not easy to configure to simulate conditions found in real-world forests, such as capturing structurally inhomogeneous

canopies, or variations in the ambient atmospheric conditions (Bannister et al., 2022). Conversely, Eulerian approaches lend themselves more easily to site observations and physical models, as well as numerical investigations. However, the

interpretation is slightly different: one calculates an average residence time of air in a flow or control volume, rather than tracking the movement of individual parcels.

Because of the challenges in calculating residence times of air from point observations, field estimates are rarely reported, meaning there is little data against which modelling estimates can be evaluated. A handful of studies have used $^{222}$Rn, a radioactive gas produced along the $\alpha$-decay chains of uranium, as an inert tracer. Because $^{222}$Rn is inert and originates in the soil, provided the ground flux is known, its concentration in the forest airspace can be used to infer a canopy ventilation rate (Martens et al., 2004; Simon et al., 2005; Trumbore et al., 1990). Trumbore et al. (1990) used $^{222}$Rn measurements to calculate

mean canopy residence times of $\leq 1$ h and 3.4–5.5 h for day- and night-time conditions, respectively, in a mature Amazon Rainforest site ($h_c \approx 30$ m). Subsequent measurements at other Amazonian locations have reported mean residence times ranging from around a minute during the day to several hours at night (Martens et al., 2004; Rummel, 2005; Simon et al., 2005). Measurements in a young ponderosa pine plantation ($h_c = 5.7$ m) in California, USA found daytime summer residence times ranging from 70–420s (Farmer and Cohen, 2008). It is possible to estimate residence times through indirect methods,

such as calculating the mean time between scalar ramps in the ejection–sweep cycle that dominates turbulent exchange between forests and the atmosphere (Katul et al., 1996; Paw U et al., 1995; Rummel et al., 2002). These methods have been used to estimate residence times of a minute or two during the day to around an hour at night (Rummel et al., 2002). However, there are no field reports of residence-time statistics beyond their mean values, which provide limited information in, for example, calculating the probability of a BVOC reacting during its passage out of a forest. Further, little is known about the influence

of even basic meteorological variables on residence times of air in forests.

Here, we report opportunistic investigations of residence times of air in the mature, broadleaf deciduous forest at the Birmingham Institute of Forest Research (BIFoR) free-air carbon dioxide enrichment (FACE) facility. The primary experiment at BIFoR FACE observes forest ecosystem behaviour under future atmospheric composition. This is achieved by using large-

scale infrastructure to elevate the $CO_2$ mixing ratio, without containment, to 150 μmol mol$^{-1}$ above ambient in several large patches of the forest (Hart et al., 2020; MacKenzie et al., 2021). BIFoR FACE is one of two 'second-generation' FACE experiments on mature, ecosystem-scale forests, the other being the 'EucFACE' experiment in an open sclerophyll forest in Australia (Drake et al., 2016). If we focus our attention on time scales of seconds to hours, over which the $CO_2$ is approximately passive, the normal course of operation of BIFoR FACE also offers a unique, daily dispersal experiment. Across three patches

of the mature woodland, the $CO_2$ mixing ratio is elevated around sunrise, held at 150 μmol mol$^{-1}$ above ambient during daylight hours, and allowed to return to ambient after sunset, when the $CO_2$ release is stopped. We use three years' data (just under fifty million observations) to investigate the effect of canopy structure and the surrounding atmospheric conditions on residence times in a mature temperate forest.

## 2 Methods

### 2.1 Site description

The BIFoR FACE facility is located in a mature deciduous broadleaf forest patch ($\approx 19$ ha) in central England, United Kingdom (latitude, longitude: 52.7996, -2.3039). The BIFoR FACE woodland is dominated by *Quercus robur* (pedunculate oak), with a dense heterogeneous understorey layer of *Corylus avellana* (hazel), *Crataegus monogyna* (common hawthorn), *Acer pseudoplanatus* (sycamore), and *Ilex aquifolium* (holly). Below the heterogeneous understorey, the woodland supports ground

flora, including *Phegopteris connectilis* (beech fern), *Rubus fruticosus* (bramble), *Hedera spp.* (ivy), *Lonicera periclymenum* (honeysuckle), and, where the canopy has been opened for access rides, various grass species (G. Platt, private communication, 2019). The BIFoR FACE woodland shows evidence of historical coppicing but it has not been managed for at least 30 years.

The largest oaks were planted in 1850. Hanging and fallen deadwood is left in place except where it poses a direct risk to human safety. The highest point of the facility is situated in the east of the forest, at around +112 m above sea level (a.s.l.) and the lowest point at the site offices and $CO_2$ storage plant, at +92 m a.s.l. The terrain below the areas of experimental interest is quite level, at +108 ± 2.7 m a.s.l. (see contour maps in MacKenzie et al. (2021)).

The BIFoR FACE facility comprises nine experimental patches of forest, which are approximately circular, with an internal radius of around 17 m (Table 1). There are three 'fumigated' (f) patches, in which infrastructure arrays maintain the $CO_2$ mixing ratio (denoted [$CO_2$] hereafter) at 150 μmol mol$^{-1}$ above ambient during daylight periods of the growing season. There are three further 'control' (c) patches, which are dosed with ambient air only, and three 'ghost' patches, which are ecologically similar to the fumigated and control patches, but do not contain any of the supporting infrastructure (Figure 1). In the fumigated arrays, premixed air/$CO_2$ is released in the upwind quadrant from perforated vent pipes, supported by 16 free-standing lattice towers (Figure 1). The wind direction and speed are updated in the FACE control program (FCP) every second, based on 20 Hz sonic anemometer measurements at the canopy top on the northernmost tower of each fumigated array (Hart et al., 2020). The forest arrays are paired, so that a control array mimics the actions of its corresponding fumigated array, but doses the forest patch with ambient air only. The pairings are numbered 1(f) and 3(c), 4(f) and 2(c), 6(f) and 5(c) (Figure 1). For more background on the BIFoR FACE facility and its operation, see Hart et al. (2020). Details of the measurements and data and tissue curation pipelines are provided in MacKenzie et al. (2021).

**Table 1: Geometries of the BIFoR FACE control (c) and fumigation (f) arrays. The internal radius is defined as the mean distance between the central tower and the inside edge of the towers supporting the perforated vent pipes.**

| Array | Infrastructure tower heights (m) | Central tower height (m) | Height of $CO_2$ sample inlet | Internal radius (m) | Research ground area (m$^2$) | Array volume (m$^3$) |
|---|---|---|---|---|---|---|
| 1(f) | 26.7 | 26.0 | 21.5 | 17 | 724 | 24,815 |
| 2(c) | 25.6 | 24.9 | 22.5 | 16 | 628 | 21,107 |
| 3(c) | 26.2 | 25.5 | 21 | 16 | 661 | 22,138 |
| 4(f) | 27.2 | 26.5 | 22 | 17 | 702 | 24,406 |
| 5(c) | 27.3 | 26.6 | 22.5 | 17 | 688 | 24,207 |
| 6(f) | 24.7 | 24.0 | 19.8 | 17 | 678 | 21,641 |

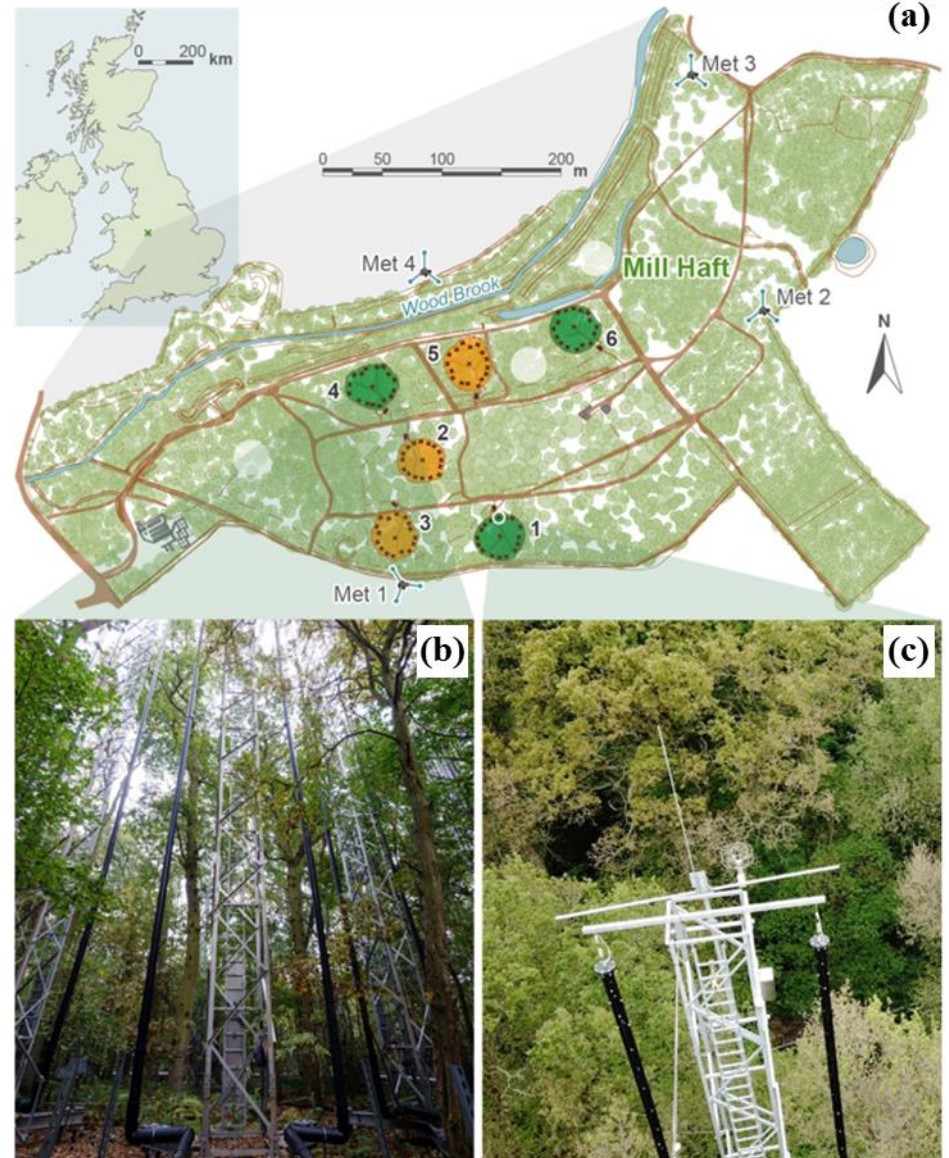

**Figure 1: (a) schematic of the Birmingham Institute of Forest Research free-air carbon dioxide enrichment (BIFoR FACE) facility (see section 2.1 for site description). The coloured circles indicate the location of the FACE arrays, with green and orange denoting the fumigated and control arrays, respectively. The grey translucent circles mark the locations of the ghost arrays. The meteorological towers on the edge of the forest are labelled Met 1–4. (b) The perforated FACE vent pipes in array 4. (c) The two-dimensional sonic anemometer in array 1 (see section 2.2.2 for details of meteorological measurements). Figure 1a © Crown copyright and database rights 2021. Ordnance Survey (100025252).**

## 2.2 Observational details

### 2.2.1 Observation period and canopy density

The FACE arrays operate up to 18 hours a day (04:30–21:30), depending on day length, and from budburst (around 1 April) to leaf fall (around 31 October). We investigate observations from 1 April–31 October in the years 2019–2021. We refer to the April fumigation period as 'leaf-off', because the dominant canopy oaks put out most of their leaves in May, and the period from 1 May to 31 October as 'leaf-on'. Together the leaf-on and leaf-off periods, as defined, make up the $CO_2$ fumigation period at BIFoR FACE. The LAI is much greater during the leaf-on period than the leaf-off period—see, for example, the hemispheric photographs in Figure 2. The LAI $\approx$ 7–8 during the leaf-on period, calculated using extensive leaf-litter measurements throughout the season. The plant area index (PAI)—the total projected plant area per unit ground area—is approximately 1–2 for the leaf-off period, however, this is only a rough estimate. Deriving PAI estimates from digital photographs, for example, is problematic in tall multi-layered forests (Yan et al., 2019) and leaf litter observations are not available. To show the broad phenological changes at BIFoR FACE, Figure 2 presents timeseries of the green chromatic

coordinate (GCC) for the investigation period. The GCC (normalised to take values from 0–1) measures the 'greenness' of the canopy from repeated digital photographs (Woebbecke et al., 1995). Figure 2 shows that the greenness of BIFoR FACE forest increases sharply towards the end of April, as the canopy oaks begin to put out their leaves, peaks in May–June, declines

slowly across the leaf-on period as the leaves mature, before declining sharply in November when the dominant oaks drop their leaves. A note of caution: although the GCC is a helpful tool to monitor seasonal canopy-scale dynamics (Toomey et al., 2015), it is not a proxy for plant-area density in multi-layered deciduous forests. For example, in Figure 2, the sharp changes in GCC in spring and autumn correspond to changes in leaf density, but the gentle decrease in GCC over the leaf-on period is not reflected by changes in canopy density (i.e., the leaves become less green over the summer, but their number remains

approximately constant).

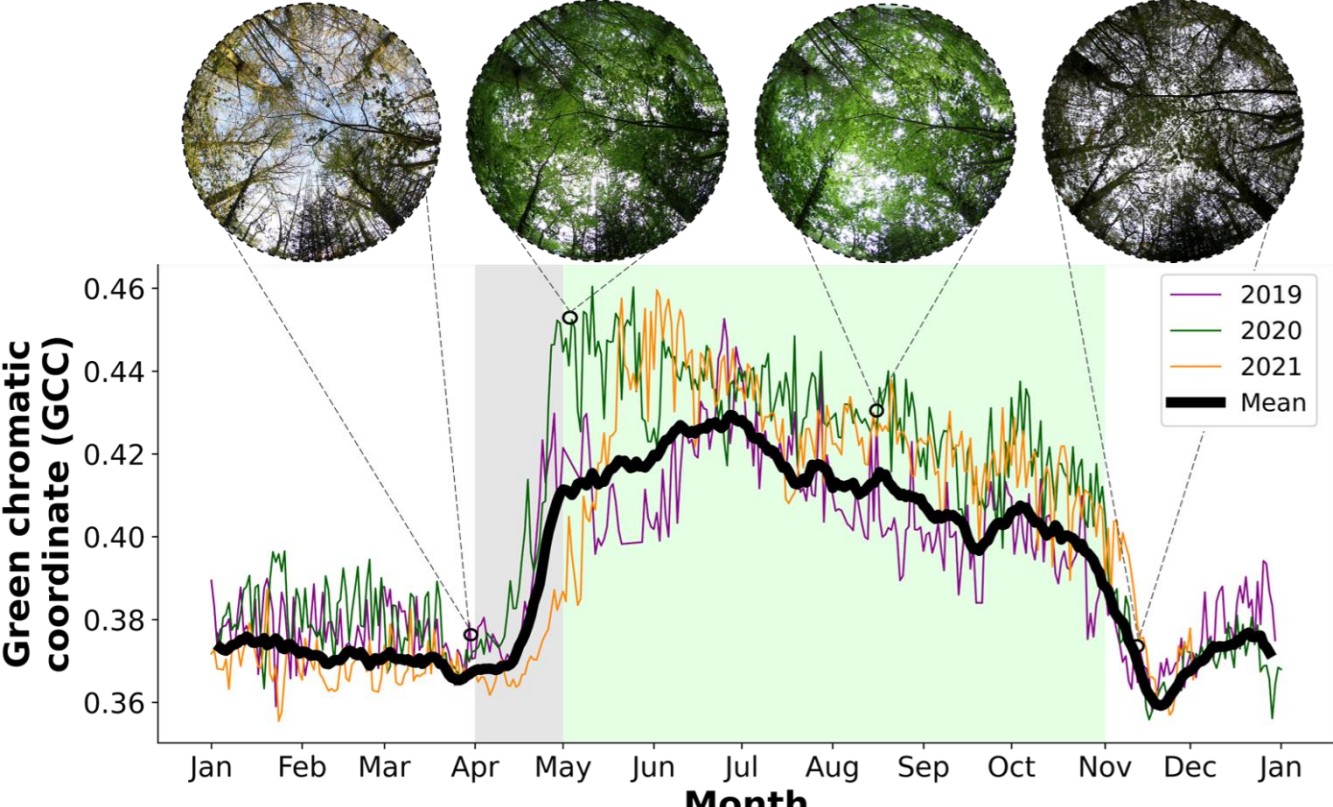

**Figure 2: Timeseries of the green chromatic coordinate (GCC) derived from PhenoCam measurements (section 2.2.1). The hemispheric photos are taken by cameras around 50 cm above the ground in array 1 (Figure 1 and site description in section 2.1). Shaded grey and green regions show the leaf-off and leaf-on periods, respectively (as defined in section 2.2.1).**


**2.2.2 Fumigation and meteorological measurements**

The FCP determines, based on solar elevation, the times at which the fumigation is started and shut down each day. Array pairings are switched on in sequence 1(f) + 3(c), 2(c) + 4(f) and 5(c) + 6(f). Wind velocities for the FCP are monitored at 1 Hz using two-dimensional sonic anemometers (WMT700, Vaisala Oyj, Vantaa, Finland), mounted at a height $z \approx 1$ m above the

canopy on the northernmost tower of each fumigated array. The FCP logs 1-min averages of the wind speed and direction, and of other variables including the air temperature, atmospheric pressure, and solar elevation. There are four meteorological masts around the edge of the forest (denoted Met 1–4, respectively; Figure 1), with three-dimensional sonic anemometers (R3-100, Gill Instruments, Lymington, UK) mounted at 25 m on each. These anemometers sampled the three-dimensional instantaneous velocity components and the speed of sound at 20 Hz throughout the entire observational period. In October 2020, three

additional three-dimensional sonic anemometers (Windmaster Pro, Gill Instruments, Lymington, UK) were added to each mast at heights of 7 m, 10 m, and 14 m, sampling the same variables at the same rate as the existing sensors. The [$CO_2$] is measured

at 1 Hz using infrared gas analysers (IRGA, LiCor 840A, LiCor Lincoln) with inlets situated in the centre of the fumigation and control arrays, just below the top of the canopy for each array (Table 1).

The FCP automatically records 1-min and 5-min averages of the 1 Hz [$CO_2$] observations. The software halts fumigation when the canopy-top 1-min average air temperature is less than 4°C because broadleaf forests uptake a negligible amount of carbon below this threshold (Larcher, 1995). Fumigation is also stopped during periods of high winds—where the 15-min average wind speed, $V$, at the canopy top exceeds 8 m s$^{-1}$—because of the high cost of maintaining the elevated [$CO_2$]. When $V < 0.4$ m s$^{-1}$, the FCP introduces $CO_2$-enriched air all around the array via alternate vent pipes, rather than in the upwind quadrant, as
under normal wind speeds. This is because advection of the enriched gas flow is ineffective at very low wind speeds.

## 2.3 Calculation of residence times

We calculate residence times from the FACE data using a mass balance approach. We treat each fumigated array as a reservoir of 'additional' $CO_2$, i.e., as a reservoir of air with a $CO_2$ mixing ratio that is elevated (e[$CO_2$]) compared with the ambient $CO_2$ mixing ratio, a[$CO_2$]. The residence time represents the average time each additional molecule of $CO_2$ spends in the fumigated
arrays before it is transported out by turbulent and advective fluxes, or is taken up by the trees and other plants. Provided we choose a time period over which the mass of the additional $CO_2$ in each fumigated array is approximately steady, the residence time can be interpreted equivalently as the time it would take to increase the $CO_2$ mixing ratio from a[$CO_2$] to e[$CO_2$] in the absence of significant sinks. First, we find the mixing ratio of the additional $CO_2$ in each fumigated array ($\chi_{eCO_2}$) during fumigation, i.e., the difference between the elevated and ambient mixing ratios:

$$\chi_{eCO_2} \ (\mu\text{mol mol}^{-1}) = \text{e[CO}_2] - \text{a[CO}_2]. \tag{3}$$

The value of $\chi_{eCO_2}$ is then used together with the ideal gas equation to calculate the mass of additional $CO_2$ in each fumigated array during fumigation:

$$M_{CO_2} = V_a M_r \chi_{eCO_2} \frac{p}{\mathcal{R}T}, \tag{4}$$

where $M_{CO_2}$ (g) is the mass of the additional $CO_2$, $V_a$ (m$^3$) is the effective volume of each fumigated array (Table 1), $M_r$ is the molar mass of $CO_2$ (g mol$^{-1}$), $p$ is the atmospheric pressure (Pa), $\mathcal{R}$ is the molar gas constant (8.314 m$^3$ Pa K$^{-1}$ mol$^{-1}$), and $T$ is the air temperature (K). For the residence time analysis across the entire study period, we treat $V_a$ as constant for each array.
However, when examining individual events such as venting in stable atmospheric conditions (section 3.7), this assumption is called into question. We define a residence time by dividing the mass of additional $CO_2$ in each array by the flow rate required to sustain it:

$$\tau = M_{CO_2}/F_{in}, \tag{5}$$

where $\tau$ (s) is the residence time and $F_{in}$ (g ($CO_2$) s$^{-1}$) is the $CO_2$ flow rate into each fumigated array from the FACE infrastructure. Eq. (5) discounts other sources of additional $CO_2$ into each fumigated array, most notably the soil fluxes ($F_{soil}$).
This is justified because $F_{in} \gg F_{soil}$ during fumigation—$F_{in} \approx 50$–550 g ($CO_2$) s$^{-1}$ in each array, compared with $F_{soil} < 0.1$ g ($CO_2$) s$^{-1}$ (Von Arnold et al., 2005).

We consider the conditions under which Eq. (5) offers a reasonable estimate of residence times. In a quasi-infinite model of a uniform forest, such as in GCF17, the only path for air parcels to leave the canopy is through vertical venting out of the top,
which we denote $F_{out(top)}$. The BIFoR FACE arrays, however, are not closed at the sides, and air parcels can also exit the arrays horizontally, i.e., there is some non-zero horizontal flux, $F_{out(hor)}$, of air out of the array. In a quasi-infinite, uniform forest, we expect $\tau = M_{CO_2}/F_{in} \approx M_{CO_2}/F_{out(top)}$. In reality, however, $\tau = M_{CO_2}/F_{in} = M_{CO_2}/(F_{out(top)} + F_{out(hor)} +$

$F_{out(sink)}$), where $F_{out(sink)}$ denotes $CO_2$ sink terms, most notably photosynthetic uptake. We do not include $F_{out(sink)}$ in our calculations below because $F_{out(sink)} \approx 0.5$–2 g $(CO_2)$ s$^{-1}$ during the day (Gardner et al., 2021), typically less than 1% of the total flux. Long-term analysis of the BIFoR FACE observations shows contamination events between the arrays are rare and mostly small (Hart et al., 2020), usually occurring at above-average wind speeds. This suggests that, although $F_{out(hor)}$ is always non-zero, it is likely small relative to $F_{out(top)}$ in conditions with weak advection. Unfortunately, horizontal fluxes in forests are difficult to measure or even estimate (Aubinet et al., 2010). Therefore, rather than trying to assign a numerical value to $F_{out(hor)}$, we identify meteorological conditions under which $F_{out(top)} \gg F_{out(hor)}$, and therefore $\tau = M_{CO_2}/F_{in} \approx M_{CO_2}/F_{out(top)}$. Figure 3 presents probability density functions of $\tau$ during the lowest 50% of wind speeds of the leaf-on period (solid black), during the highest 25% of wind speeds of the leaf-on period (dashed), and GCF17's model in Eq. (1) (navy). Because the mean horizontal wind speed decays exponentially with height in forest canopies (Finnigan, 2000), in weak-wind conditions (here, below $\approx 2$ m s$^{-1}$ at $z = 25$ m), the wind speed at the height of the fumigation is very low. Advection is therefore likely small compared with the turbulent exchange driven by the large eddies around the top of the canopy (Finnigan, 2000; Raupach et al., 1996). The PDF for GCF17 takes $K_{eq} = 1.2$ m$^2$ s$^{-1}$, calculated using Eqs. (A1–3) in Appendix A, $h_c = 25$ m, and $z_{rel} = 15$ m. The fumigation at BIFoR FACE is not uniformly vertically distributed (see section 2.4 below). It is therefore difficult to determine a single release height, $z_{rel}$, as used for the Lagrangian parcels in GCF17. We found $z_{rel} \approx 15$ m gave the closest agreement between our results and GCF17.

We use a 5-min averaging period for the residence time calculations in Equations (3–5) and Reynolds averaging of the meteorological tower observations below (section 2.4). Sensitivity testing on high-resolution velocity measurements showed this to be the most appropriate period to capture the significant turbulent structures at this structurally heterogeneous site, while being long enough so that $\chi_{CO_2}$ and $F_{in}$ were approximately steady. In mature forests, whose largest eddies scale with the mean height of the canopy $h_c$ (Bannister et al., 2022; Finnigan, 2000; Raupach et al., 1996), the canopy turnover time $\tau_c \approx h_c/u_* \approx$ 30–90 s, where $u_*$ is the friction velocity measured at $z = h_c$ (section 2.5 describes the calculation of $u_*$ in this paper). This averaging period therefore corresponds to 5–10 cycles of the dominant turbulent eddies and the statistics of the residence time calculations were not qualitatively altered using averaging periods of up to 1 hr.

Figure 3 shows, at low wind speeds, our method generates PDFs of $\tau$ in reasonably close agreement to GCF17, especially given the very different assumptions used to calculate each PDF. Under these conditions, the one notable deviation between our results and GCF17's theory is in the right tails of the PDFs (Figure 3b), which we discuss further in section 3.6. In the strongest winds, however, the limited diameters of the BIFoR FACE arrays constrains our method. In these conditions, the mostly small values of $\tau$—visible in the sharp peak of the PDF in Figure 3a and steep decay of the right tail in Figure 3b—indicate that $F_{in}$ has increased, and therefore the flux out has increased. Comparisons with GCF17 suggest this is predominantly due to an increase in the horizontal component $F_{out(hor)}$, which is difficult to approximate in our finite-size arrays. Our residence-time calculations below therefore include only observations during the lower half of wind speeds (varying the percentile cut-off between 40–60% does not qualitatively affect our results). We discuss the implications of stronger winds on $\tau$ in sections 3.3 and 3.6.

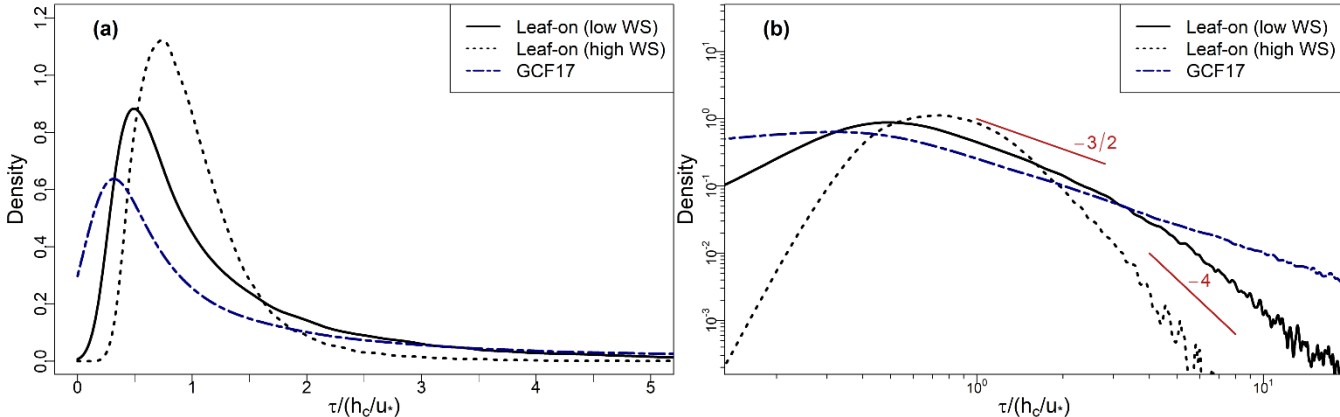

Figure 3: (a) Linear- and (b) logarithmic-scale PDFs of τ, as defined in section 2.3, from BIFoR FACE during the lowest 50% (solid black) and highest 25% (black dashed) of wind speeds of the leaf-on period. GCF17's model in Eq. (1) is shown with the navy-blue dot-dash line. In (b), slopes of -3/2 and -4 are shown for reference. The slope of -3/2 is the power-law decay from GCF17, Eq. (1), and -4 is an arbitrary value to show the steeper decay of the right tail of the PDF of τ in strong winds. Values of $\tau$ normalised by the canopy turnover time $h_c/u_*$ (section 2.4).

## 2.4 Data processing

We discarded observations for dates on which at least one of the fumigation arrays was switched off for more than two hours, or switched on and off more than once during the normal fumigation period. These temporary shutdowns were usually for maintenance work, or during periods of exceptionally high winds (which we discarded in any case according to section 2.3). This cautious filtering threshold ensures the residence time calculations focus on periods during which the fumigation was steady, rather than when the FACE infrastructure was operating at high flow rates to increase the [CO₂] following shutdown. We also discarded dates on which observations were available from neither Met 1 nor Met 4 (see Figure 1). The filtering process left 530 observation days (78 in leaf-off and 452 in leaf-on) from a total of 642 (90 in leaf-off and 552 in leaf-on). To avoid erroneous values of $\tau$, we discarded entries where: (i) $F_{in} < 1$ g (CO₂) s$^{-1}$; and (ii) values of $M_{CO_2}$ lay outside the range $\overline{M_{CO_2}} \pm 4\sigma(M_{CO_2})$, where $\sigma$ is the standard deviation and the overbar denotes the mean. Steps (i) and (ii) together discarded less than 0.3% of the data.

To aid comparisons with previous reports, we highlight two features of the fumigation at BIFoR FACE. First, the fumigation is only conducted when the trees are likely to be photosynthesising, i.e., during the daytime (with an hour or so of fumigation either side of sunrise and sunset) of the UK growing season, which is taken as 1 April–31 October. We therefore emphasise our estimates here are of residence times of air during the daytime of the northern temperate spring, summer, and autumn. The BIFoR FACE infrastructure is configured to prioritise fumigation to the regions of the canopy with the most photosynthesising leaves. The e[CO₂] outlet ports on the fumigation towers are therefore most numerous and spaced closest together between heights of 14–25 m, where the bulk of the oak canopy is located, and below around 10 m, in the coppice and understorey layers of the forest (Hart et al., 2020).

We use three measures of statistical variability in this paper: the standard deviation, the interquartile range (IQR), and the median absolute deviation, $D_{med} = \text{median}(|x_i - \tilde{x}|)$ for a univariate set $x_1, x_2, ..., x_n$, with $\tilde{x}$ the set's median. The $D_{med}$ is helpful when considering the spread of observations with highly skewed distributions, as is the case here. For highly skewed distributions, the more familiar standard deviation overweighs the influence of (absolutely) large values in the observations (Jobst and Zenios, 2001). In such cases, the $D_{med}$ provides a less volatile and more representative measure of a sample's deviation. However, this paper retains the standard deviation to aid comparison to other works, because it is more commonly reported than measures such as the $D_{med}$. We report the IQR because it is familiar and provides a robust measure of the spread of the middle 50% of a dataset (but contains no direct information on the tail behaviour).

## 2.5 Notation and meteorological tower calculations

We use right-handed Cartesian coordinates throughout this paper. We denote $\mathbf{x} = (x, y, z)$, the velocity components $u, v, w$ (using the meteorological convention that positive $u$ and $v$ values indicate westerly and southerly flow, respectively), and time as $t$. For a quantity $\phi(\mathbf{x}, t)$, a double overbar denotes the time average and the prime denotes the deviations from that average, which we refer to as the 'turbulent quantities', i.e. $\phi(\mathbf{x}, t) = \bar{\bar{\phi}}(\mathbf{x}) + \phi'(\mathbf{x}, t)$. The double overbar is used instead of the conventional single overbar to distinguish the time averages from the descriptive statistics elsewhere in the paper. The turbulence kinetic energy (TKE) per unit mass $= \frac{1}{2}(\overline{u'^2} + \overline{v'^2} + \overline{w'^2})$. The friction velocity $u_* = \left(\overline{u'w'}^2 + \overline{v'w'}^2\right)^{\frac{1}{4}}$ is a scaling variable that is most meaningfully defined in the inertial sublayer of the atmosphere (Monin and Obukhov, 1954). However, it is often used as a shorthand for turbulence elsewhere in the atmospheric surface layer, with higher values indicating more turbulent conditions. The Obukhov length, $L$, is calculated as

$$L = \frac{-\bar{\bar{T}}_s u_*^3}{\kappa g \overline{w'T_s'}},$$

(6)

where $\kappa = 0.4$ is the von Kármán constant, $g$ is the acceleration due to gravity, and $T_s$ is the sonic air temperature, which is a good approximation of the virtual potential temperature (Kaimal and Gaynor, 1991). The values of $L$, $u_*$ and the TKE are calculated from 20 Hz observations at $z \approx 22$ m $\approx h_c$ on Met 4 preferentially, because it lies at the downstream edge of the forest in the direction of the prevailing wind. On dates for which Met 4 observations were unavailable, the observations were taken from Met 1 (Met 4 and Met 1 account for 512 and 18 days, respectively, of the 530 total).

## 2.6 Stability classes

To analyse the dependence of $\tau$ on atmospheric stability, we define three broad stability classes following the approach in Mahrt et al. (1998) and Dupont and Patton (2012). Our stability regimes are defined at $z \approx h_c$ according to the behaviour of the kinematic fluxes of temperature $\overline{w'T_s'}$ and momentum (via $u_*$), as a function of the stability parameter $h_c/L$ from Eq. (6).

- Near-neutral (NN): $-0.005 \leq h_c/L < 0.003$. In this regime, the momentum flux is significant, but the temperature flux is negligible.
- Stable: $3 \leq h_c/L < 20$. This regime occurs mostly in light winds, often on cloudy mornings or shortly before fumigation shutdown in the evening. The momentum flux is small. Intermittent turbulence is a major component of turbulent exchange (Mahrt, 2014).
- Unstable: $-20 \leq h_c/L < -1$. This regime mostly occurs during the day, especially in clear-sky conditions. This regime is characterized by a large temperature flux and, usually, small $u_*$ values associated with light winds.

These thresholds are not universal and are site and study specific. We define only three broad stability classes and adopt unusually demanding thresholds to define them. This is because (i) fumigation is carried out mostly during the day, so we have limited opportunity to investigate transitory sub-regimes, which typically occur in the early morning and late evening; and (ii) the observations used to calculate $L$ are not taken at exactly the same location as the observations used to calculate $\tau$, so we prefer to exclude potentially misleading marginal cases.

# 3 Results and Discussion

## 3.1 Wind conditions at BIFoR FACE

Figure 4 presents wind roses for the 2019–2021 fumigation period at BIFoR FACE across the period as a whole (Figure 4a, c)

and for the observations used in the residence-time calculations, i.e., the lowest 50% of wind speeds across the whole observation period (Figure 4b, d). The wind speeds are generally low compared with observations from most meteorological stations because the wind measurements at BIFoR FACE are measured around the tops of the trees of each array, whereas meteorological stations are typically located away from large obstacles. The wind speeds were generally higher in the leaf-off period than the leaf-on. For example, 39% of 5-min averages were > 2.5 m s$^{-1}$ for the leaf-off period, compared with 27% for

leaf-on. The prevailing wind direction around BIFoR FACE is south-westerly, as is typical for most of the UK. However, the wind direction in the UK is highly variable in April (leaf-off) and the wind direction around BIFoR FACE was predominantly easterly during the leaf-off period 2019–2021 (Figure 4a, b). Predominantly easterly winds are unusual in the UK, but these observations match the local synoptic conditions over the same period. Our leaf-off period is much shorter than the leaf-on period and is therefore more susceptible to isolated meteorological events.

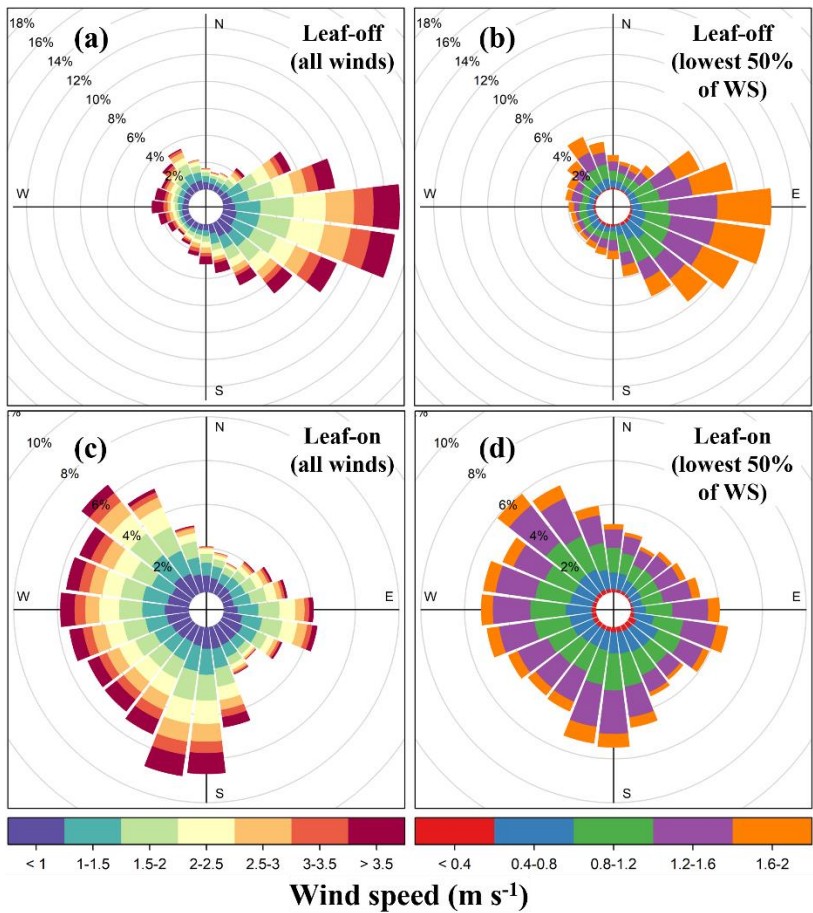

**Figure 4: Wind roses for BIFoR FACE during the 2019–2021 leaf-off (a, b) and leaf-on periods (c, d) (see section 2.1 for site description, and 2.2 for observation details). The wind roses are calculated on 5-min averages of sonic measurements in array 1. (a, c) show wind roses across all wind conditions; (b, d) show wind roses for the lowest 50% of wind speeds, used in the residence-time calculations (section 2.3). The wind roses for the other fumigation arrays are very similar and are omitted to avoid repetition. Note**

**the change of scale between (a, c) and (b, d).**

## 3.2 Basic distributions of $\tau$ values

Figure 5 presents probability density functions (PDFs) and reports descriptive statistics of the residence times for the leaf-off ($\tau_{off}$) and the leaf-on ($\tau_{on}$) periods. The overbar and overtilde notation refer to the mean and median values, respectively. The modal values of $\tau < \tilde{\tau} < \bar{\tau}$ for each period, which is typical but not diagnostic (von Hippel, 2005) of positively skewed unimodal

distributions. Longer residence times are relatively less common during the leaf-off period than leaf-on, as indicated by the

shift to the left of the $\tau_{off}$ PDF compared with the $\tau_{on}$ PDF. For example, 57% of $\tau_{on}$ observations are greater than 60 s, compared with only 24% of $\tau_{off}$ values. The $\tau_{on}$ values are more dispersed than the $\tau_{off}$ values. For example, the interquartile range (IQR) for $\tau_{on}$ is over twice that of $\tau_{off}$, and $D_{med}(\tau_{on}) > D_{med}(\tau_{off})$, where $D_{med}$ is the median absolute deviation.

In Figure 5b, both the leaf-off and leaf-on PDFs show clear modal values, followed by a region over which the decay exhibits almost power-law behaviour, followed by steeper decay in the tails.

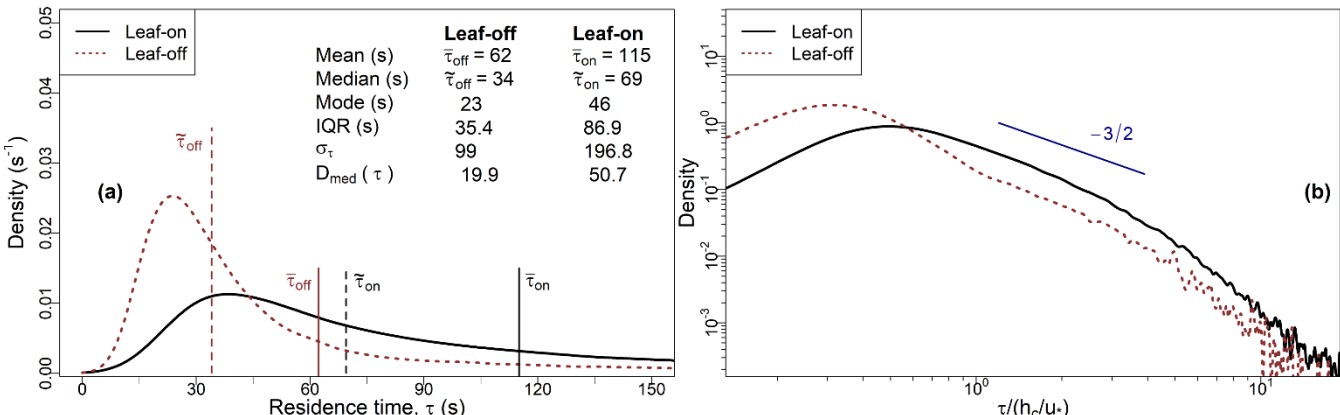

**Figure 5: (a) PDFs and statistics of the residence times for the leaf-on and leaf-off periods (see section 2.2 for observation details, and 2.3 for calculations of the residence time, $\tau$). Solid and dashed vertical lines mark the mean and median values for each period, respectively. The mode for each period is taken as the value at which the PDFs attain their maximum densities. (b) As for (a), with PDFs presented on log-log axes with $\tau$ normalised by the canopy turnover time, $h_c/u_*$ (section 2.4). The black line is the same as in Figure 3, although (a) presents dimensional information whereas Figure 3a presents the normalised PDF.**

A detailed consideration of in-canopy chemical reactions is outside the scope of this investigation. However, it is illustrative to consider these findings in the context of chemically reactive tracers, such as BVOCs, while keeping in mind that our results pertain to air in the canopy during the daytime. Our results suggest that molecules are unlikely to have time to react within the forest unless their chemical reaction time scale $\tau_{chem}$ is in the order of a few minutes or less. As an example, isoprene has a $\tau_{chem}$ of around 300–5000s in temperate oak canopies (Karl et al., 2013). Our results give a Damköhler number $Da = \tau/\tau_{chem} \approx 0.02$–$0.3$ for isoprene during the leaf-on season, in near-neutral atmospheric daytime conditions. These $Da$ values suggest a degree of conversion of up to around 20%, meaning that most isoprene will not have time to react before it is vented from the canopy. However, the degree of conversion for many chemical species increases rapidly as $\tau_{chem}$ approaches $\tau$ (GCF17; Karl et al., 2013). Therefore, in conditions where longer residence times are more likely, such as in stable atmospheric conditions (see section 3.3 below), the degree of conversion for short-lived species may increase superlinearly relative to the increase in $\tau$ (see section 4.4 GCF17 for a discussion of this topic in a Lagrangian modelling context).

### 3.3 Dependence of $\tau$ on $u_*$ and atmospheric stability

### 3.3.1 Dependence of $\tau$ on $u_*$

Figure 6 presents combined scatter and density plots showing the variation of $\tau$ with $u_*$ in the (a) leaf-off and (b) leaf-on periods. The warmer colours indicate regions of higher density. The colour scale is normalised to account for the different sample sizes in the two periods. Figure 6 shows, over both periods, the residence times decrease with increasing values of $u_*$. This accords with intuition that canopy residence times should progressively reduce with increasing turbulence. Most notably, (a) and (b) regress to gradients of $\approx -1$ ($-0.93$ and $-0.95$, respectively), which indicates that, as a first approximation, the effect of turbulence levels on the residence times is given by $\tau \propto u_*^{-1}$, as proposed by GCF17. It is worth qualifying this point a little. Because our $u_*$ values are derived from a single measurement location whereas our $\tau$ values in three nearby locations within 300 m (Figure 1), this argument assumes a state of "moving equilibrium" (Yaglom, 1979), in which $u_*$ varies slowly in the $x, y$ plane, with $u_*$ measured at $z = h_c$ serving as a local velocity scale. This assumption has not been assessed in patchy forests such as that at BIFoR FACE, whose structure varies strongly in the $x, y$ plane, likely challenging the assumption that

$u_*$ is approximately constant. Further, our results do not account for the effect of strong winds on $\tau$, which to our knowledge remains untested.

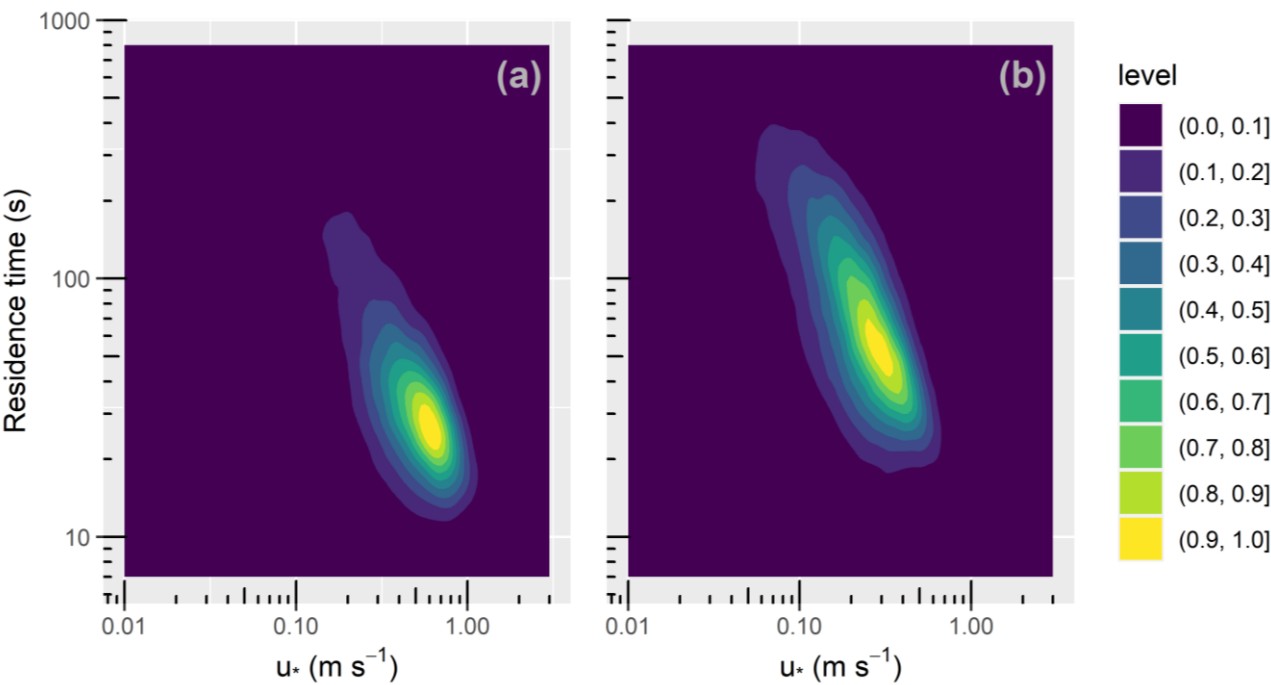

**Figure 6: Two-dimensional density plots, showing the variation of the residence time $\tau$ (section 2.3) with the friction velocity $u_*$**
430 **(section 2.5) for the leaf-off (a) and leaf-on (b) periods (section 2.2 for observation details). The colour scale is normalised to account for the different sample sizes in the two periods. 'Level' in the colour scale refers to the density of each bin, normalised by the peak density for each observational period.**

### 3.3.2 Dependence of $\tau$ on atmospheric stability

Figure 7a shows box-whisker plots of $\tau$ for the three stability classes defined in section 2.6, and Table 2 summarises their basic
435 statistics. Figures 7b, c present PDFs for the three regimes during the leaf-on period (those for leaf off are similar and are included in Figure A1 in Appendix B). The values of $\tau$ in Figure 7 are normalised by $\tau_c = h_c/u_*$ for each class to minimise the more trivial dependence of $\tau$ on $u_*$, because $u_*$ varies between the classes. However, Table 2 presents the statistics in dimensional form for easier interpretation.

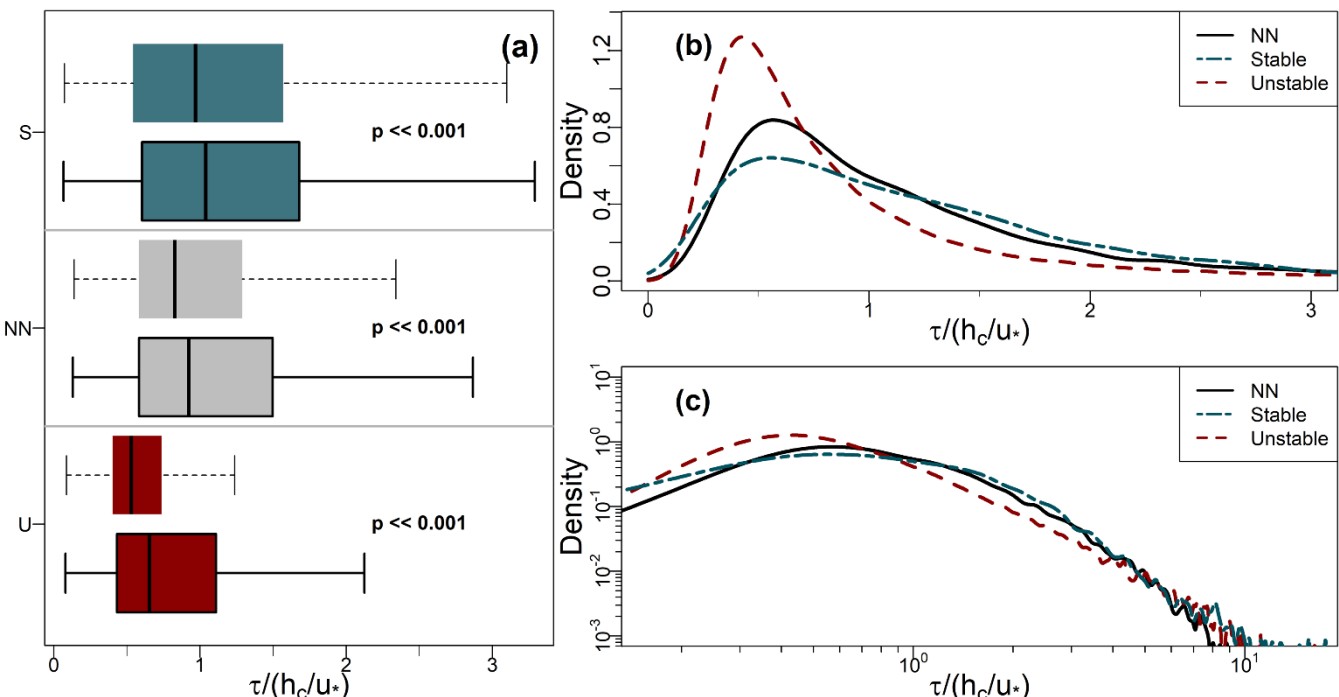

**Figure 7: Statistics of normalised residence times (section 2.3) binned by the stability classes defined in section 2.6. (a) Box-whisker plots of normalised residence times for stable (S), near-neutral (NN), and unstable (U) conditions. Boxes with dashed whiskers and no border show leaf-off values; boxes with solid whiskers and borders show leaf-on (see section 2.2 for observation details). Solid vertical lines indicate median values. Width of the boxes shows the IQR. Lower and upper whiskers respectively indicate the 25th percentile $-1.5 \times$ IQR and 75th percentile $+1.5 \times$ IQR. (b) and (c) PDFs of residence times for the leaf-on period, plotted on linear and logarithmic axes (base 10), respectively.**

**Table 2: Descriptive statistics of $\tau$ values (section 2.3) for the leaf-on and leaf-off periods binned into three stability regimes. All values in seconds rather than normalised units (other than the skewness, which has no units). The symbols $\bar{\tau}$, $\sigma_\tau$, $\tilde{\tau}$, and $D_{med}(\tau)$ denote the mean, standard deviation, median, and median absolute deviation, respectively (see sections 2.4 and 3.2 for an overview of these statistics).**

| | Stable | | NN | | Unstable | |
|---|---|---|---|---|---|---|
| | **Leaf-on** | **Leaf-off** | **Leaf-on** | **Leaf-off** | **Leaf-on** | **Leaf-off** |
| | n = 11,291 | n = 1,556 | n = 10,668 | n = 2,865 | n = 32,001 | n = 8,846 |
| $\bar{\tau}$ | 229 | 155 | 100 | 64 | 89 | 37 |
| $\sigma_\tau$ | 319 | 198 | 172 | 83 | 154 | 45 |
| $\tilde{\tau}$ | 169 | 117 | 73 | 45 | 55 | 27 |
| $D_{med}(\tau)$ | 121 | 85 | 47 | 24 | 34 | 12 |
| **IQR** | 174 | 124 | 72 | 38 | 57 | 17 |
| **Skewness** | 12.4 | 13.7 | 30.1 | 15.4 | 21.2 | 9.5 |

Residence times increase with increasing stability, as does the spread in their values. These differences are significant, both between the growing periods and between the stability classes in each period ($p \ll 0.001$ using the Mann–Whitney–Wilcoxon test). In unstable conditions, long residence times are much less common than they are in the NN or stable regimes. For example, in Figures 7b, and 7c, the right tails of the unstable PDF are lighter than those for NN and the stable regime. The distributions of $\tau$ remain positively skewed for each stability class (Table 2, and the right whiskers are longer than the left in Figure 7a). These general patterns are not sensitive to the exact thresholds of $h_c/L$ used to bin the data. Changes in the turbulence structure around the forest likely account for the main differences in the distributions of $\tau$ across the three stability classes. In NN conditions, shear generated eddies around the tops of the trees dominate turbulent exchange (Bannister et al., 2022; Brunet, 2020; Finnigan, 2000). However, as stability decreases from NN to free convection in the unstable regime, the dominant turbulent structures around the forest transition from shear-layer vortices to thermal plumes. These thermal plumes have typical length scales several times larger than shear-layer vortices (Patton et al., 2016), which could result in more vigorous mixing in unstable conditions than NN, resulting in the smaller $\tau$ values seen for the former than the latter (Figure 7). Conversely, in stable conditions, in-canopy turbulence is much weaker and more intermittent than in neutral or unstable conditions, reflected in (i) the larger average values of $\tau$ for the stable regime than the NN or the unstable regimes; and (ii) a greater likelihood of long $\tau$ values, when air remains within the canopy until it is vented by infrequent, intermittent turbulence, as reflected in the heavy tails of the stable PDFs. Section 3.7 discusses intermittent venting in stable atmospheric conditions in more detail. For the NN and unstable regimes, $\tau \propto u_*^{-1}$, but $\tau \propto u_*^{-0.8}$ in stable conditions.

**3.4 Dependence of $\tau$ on wind direction**

Figure 8 presents polar plots showing percentiles in $\tau$ values with wind direction. The values of $\tau$ are not completely symmetrically distributed with regards to wind direction. This is unsurprising because the BIFoR FACE forest is a complex, mature woodland, within which the species composition, tree age, and stand structure varies. Array 1 provides the clearest example of the heterogeneity in that the residence times are noticeably lower when the wind direction is from the south and south-east (Figures 8a and 8d). This is because array 1 is located at the southern edge of the forest (Figure 1) and therefore vulnerable to edge effects from southerly winds. However, in most mature forests, structural heterogeneity means that point observations are never likely to be entirely neutral with respect to wind direction, even when edges are accounted for. For

example, the closest edge to arrays 4 and 6 is to the north (Figure 1). But arrays 4 and 6 are relatively more exposed to south-westerly and southerly winds, respectively, because the trees abutting the arrays in those directions are slightly shorter than those to the north. These patterns do not materially change with the time of day or atmospheric stability (during daylight hours, for which we have observations). Long-term analysis of the BIFoR FACE observations show contamination of the airspace in control arrays by the e[$CO_2$] air from fumigation arrays is rare, but occurs most frequently when the control array is directly downstream of a fumigation array relative to the mean wind direction (Hart et al., 2020).

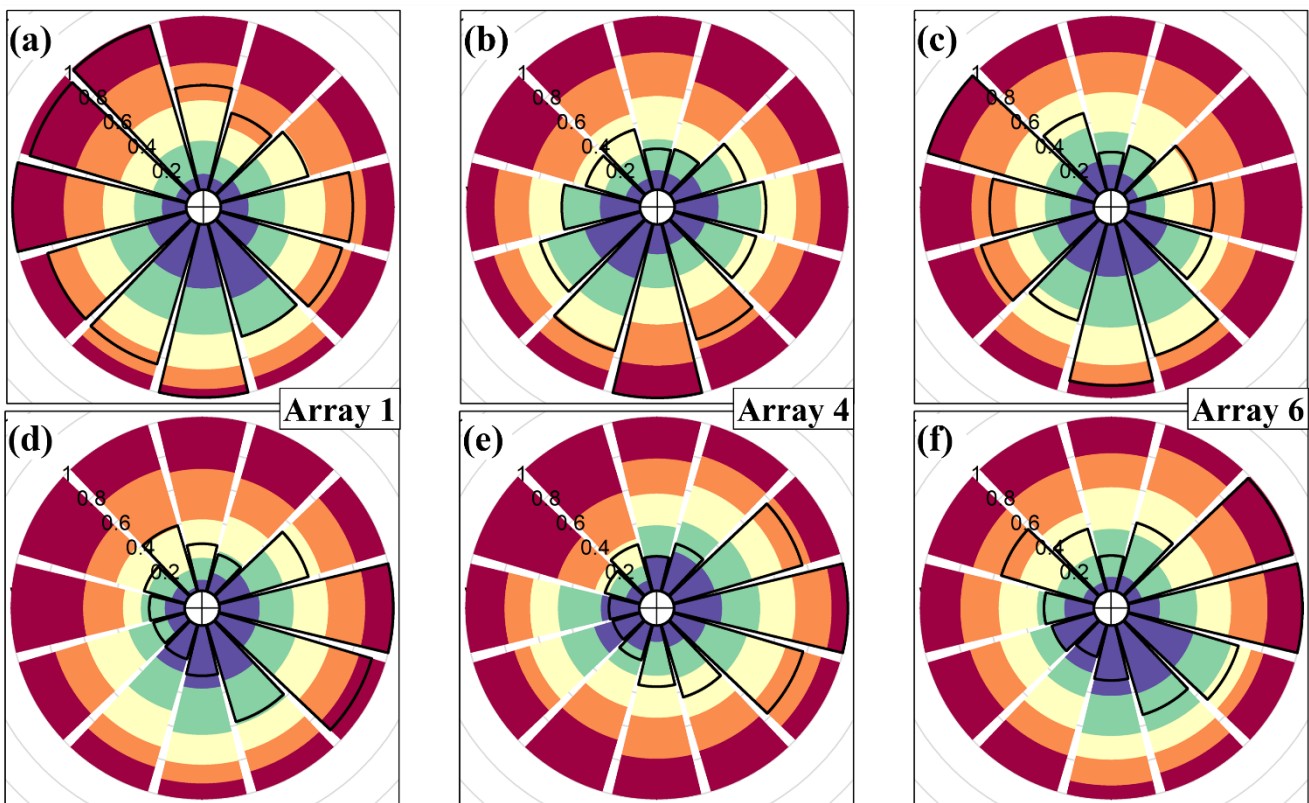

**Figure 8: Residence-time (see section 2.3) quintiles by wind direction for the leaf-on (a–c) and leaf-off (d–f) periods (section 2.2 for observation details). (a, d) Array 1; (b, e) Array 4; and (c, f) Array 6 (Figure 1 and section 2.1). The colours indicate the proportion of $\tau$ values within each quintile, increasing from blue (lowest 20% of $\tau$ values across the whole site) to red (highest 20% of $\tau$ values across the whole site). For example, (a) shows that for southerly winds, a higher proportion of $\tau$ values are in the first and second quintiles (more blue and green in the southerly wind sectors) and a lower proportion are in the fifth quintiles (less red in the southerly wind sectors). In other words, for Array 1, lower $\tau$ values are more common for southerly winds. The solid black line shows the relative frequency of each wind sector across the whole leaf-on and leaf-off measurement periods, with the scale 0–1 indicated by the radial numbering.**

No systematic differences or symmetries are apparent between the southern-edge array (array 1) and the northern-edge arrays (4 and 6). Because wind directional effects are so site and climate specific, it is difficult to generalise these results other than to say, where possible, observational campaigns of forest-atmosphere exchange in patchy landscapes should include at least two measurement locations, one deep in the forest, and one near any edges, especially in the direction of the prevailing wind. Forest edges experience different wind conditions, chemistry, microclimates to forest interiors (Bonn et al., 2014; Schmidt et al., 2017). It is important not to dismiss forest-edge processes as unrepresentative, however, because edges comprise the majority of the forested area in many parts of the world (Bannister et al., 2022).

### 3.5 Seasonal (leaf-on/leaf-off) differences in $\tau$

As indicated by the descriptive statistics in section 3.2, the forest is more ventilated when the trees are not in leaf. Taking the distributions of $\tau$ across the entire fumigation period, the values of $\tau_{on}$ are significantly higher than the values of $\tau_{off}$, with $p \ll 0.001$ using both the $t$–test and the Mann–Whitney–Wilcoxon test. Figure 8 shows that, for a given percentile, $\tau_{on} < \tau_{off}$ for most wind directions, particularly in arrays 1 and 4, which are slightly less sheltered than array 6. Figure 7 and Table 2

shows the average values of $\tau_{off} < \tau_{on}$ for the three stability classes we defined, with the distributions remaining significantly different ($p \ll 0.001$). The spread in the $\tau_{on}$ values is higher than in $\tau_{off}$ across the entire fumigation period, and in unstable

conditions. However, for the NN and stable regimes, the variability in the $\tau$ values is quite similar between the two periods.

**3.6 Comparison with published residence-time values**

Recalling the set-up of the FACE operations, described above, our estimates are most comparable to the daytime residence times of air parcels released from approximately the upper two-thirds of the canopy, $z/h_c > 1/3$. With these considerations in mind, our calculated residence times fall within the range of modelled median values of tens of seconds to a few minutes

(Fuentes et al., 2007; Gerken et al., 2017; Strong et al., 2004). There are few reported observational estimates of residence times, and none derived from measurements in ecosystems similar to the BIFoR FACE forest. To the extent a comparison is meaningful, our calculated residence times are within the range of reported field estimates e.g., mean values of a minute or two during the growing season (Farmer and Cohen, 2008; Martens et al., 2004).

Our results agree with existing modelling studies that the distributions of residence times are strongly positively skewed and in certain conditions—e.g., in stable conditions (Figure 7) or for parcels travelling from near the ground (GCF17; Strong et al., 2004)—can be widely dispersed with quite heavy tails. For these situations, average values cannot be said to be 'representative', and it is preferable to be able to estimate distributions rather than single values. GCF17's model in Eq. (1) is appealing because it allows the distribution to be estimated from a small set of variables, making it suitable for deployment in

large-scale models. The eddy diffusivity $K_{eq}$ can be partially tuned to account for the forest structure and wind conditions. However, although GCF17's model generates modal values similar to those we observed, the right tails of the distributions differ between the two studies. For example, GCF17 predicts around 20% of air parcels have residence times of five minutes or more whereas, in our leaf-on data, the proportion is closer to 6%. Some of the discrepancy between our observations and GCF17's model likely results from our underestimation of $\tau$ because of the finite-size arrays used in the mass balance

calculations in Eq. (5). However, given that GCF17's model and our results diverge even in low winds, when advection is negligible and turbulence is weak, this factor is unlikely to be the only relevant difference. Indeed, the tails of GCF17's own LES-generated PDFs appear to decay faster than the $-3/2$ power law predicted by analytical model in Eq. (1)—especially for parcels released higher in the canopy—suggesting that Eq. (1) may overpredict the likelihood of long residence times. However, we are cautious in drawing firm conclusions here because the definition of a residence time differs slightly between

GCF17 and our study. GCF17 calculated statistics on individually tracked Lagrangian 'parcels' within an LES flow, whereas we calculate a mean residence time of air within a control volume, over a five-minute period.

To proceed, it is helpful to examine the assumptions of previous approaches. The eddy-diffusivity closure assumptions used to formulate Eq. (1) are most realistic when the length and time scales of the transport mechanism are smaller than the scale

of the gradients in the measured quantities (Corrsin, 1975). Cava et al. (2006) show this condition is most likely to be satisfied when the sum of the turbulent transport and buoyant production terms in the transport equations is small compared with the gradient in the measured quantity. In forests and other vegetation canopies in neutral conditions, this is a reasonable assumption below around $z/h_c = 1/2$, especially when considering quantities with strong gradients, such as fertilizer (Bash et al., 2010) or fungal spores. However, in forest crowns in neutral conditions, turbulent exchange is dominated by eddies with diameters

that scale with $h_c$ (Brunet, 2020; Finnigan, 2000; Raupach et al., 1996). These eddies create significant turbulent transport, meaning that the eddy-diffusivity model underestimates turbulent forest-atmosphere exchange in the upper canopy and therefore overestimates residence times. As mentioned above, using LES to resolve the flow—as in GCF17—partially navigates this issue, because the turbulence parametrisation does not need to be specified a priori (although the ability of LES to resolve the flow in forests is by no means perfect). However, LES models of forests (including that in GCF17) often envisage

a horizontally homogeneous, quasi-infinite forest, in which the only path of exit for air parcels is via turbulent exchange at the top of the canopy (Bannister et al., 2022). Real forests typically comprise a patchwork of gaps and clearings at all heights, caused by disease, tree senescence, human activities, and wind throw. These openings offer air parcels additional routes to exit forests, such as via advection across edges or through the regions of strong turbulent fluxes that form in patchy forest crowns. In hilly terrain, flow-separation regions in the lee of hills can create chimney-like pathways for air parcels to leave the forest,

particularly for parcels moving from near the ground (Bannister et al., 2022; Chen et al., 2019). The likely net effect of these additional pathways is to reduce the incidence of very long residence times, particularly in forests with extensive edge regions and patchy structures.

   Here we adapt GCF17's model to reduce the overprediction of large $\tau$ values while keeping it simple enough to be deployed

in regional or global models, for which information on the canopy structure and the flow of air is typically limited. First, we observe that Eq. (1) is a special case of the inverse-gamma distribution, the general form of which is

$$p(\tau; \alpha, \beta) = \frac{\beta^{\alpha}}{\Gamma(\alpha)} \tau^{-(\alpha+1)} \exp[-\beta/\tau]\,; \tau > 0, \tag{7}$$

where $\Gamma(\cdot)$ is the gamma function and $\alpha$ and $\beta$ are, respectively, shape and scale parameters ($\beta$ is the rate parameter from the point of view of the gamma distribution). Taking $\alpha = \frac{1}{2}$ and $\beta = \tau_{turb} = (h_c - z_{rel})^2/4K_{eq}$ in Eq. (7) gives Eq. (1). The value of $\beta$ is relatively more influential at lower values of $\tau$, whereas $\alpha$ determines the distribution's dominant behaviour for

large $\tau$. In forest crowns, turbulent exchange scales with the canopy turnover timescale $\tau_c = h_c/u_*$, which we use as our value for $\beta$. The value of $\alpha$ then determines the shape of the distribution, particularly at large $\tau$ values. We find $\alpha = 1.4$–$1.8$ fits our observations better than using $\alpha = \frac{1}{2}$, as in GCF17 (Figure 9). The main effect of the larger $\alpha$ value is to reduce the probability of very long residence times, as evidenced by the roll-off of our PDFs from GCF17 at large $\tau$ values in Figure 9. A helpful by-product is that, for $\alpha > 1$ in Eq. (7), the mean values of $\tau$ become formally defined as $\bar{\tau} = \beta/(\alpha - 1)$ (the mean is undefined

for $\alpha < 1$). For our data, $\tau_{c(on)} = 78$ s and $\tau_{c(off)} = 54$ s. Taking $\alpha = 1.6$ and $1.8$ as rough estimates for the leaf-on and leaf-off periods, respectively, gives $\bar{\tau}_{on} = 130$ s and $\bar{\tau}_{off} = 68$ s, close to the values $\bar{\tau}_{on} = 115$ s and $\bar{\tau}_{off} = 62$ s calculated directly on our data.

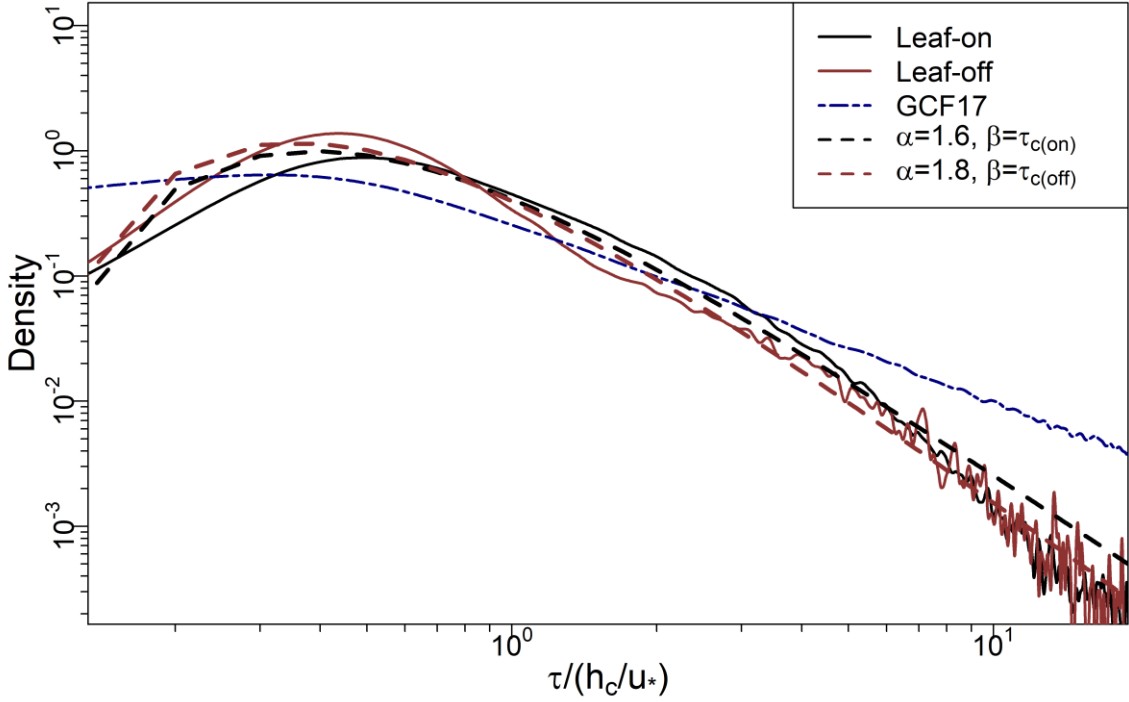

**Figure 9: Solid black and red lines show PDFs of $\tau$ (section 2.3) from BIFoR FACE (2.1 and Figure 1) during the leaf-on and leaf-**
**off periods, respectively (section 2.2). Dashed lines show PDF estimates on the BIFoR FACE observations using Equation (7). Dot-**

dash navy line shows PDF from GCF17 in Eq. (1). $\tau_{c(on)}$ and $\tau_{c(off)}$ denote $\tau_c = h_c/u_*$ for the leaf-on and leaf-off periods (section 2.4). All $\tau$ values normalised by $\tau_c = h_c/u_*$.

Inverse-gamma distributions are flexible and can fit observations from a variety of processes, without always reflecting the underlying physical mechanisms. However, surface renewal theory (SRT) (Danckwerts, 1951) offers a compelling analogy that warrants further testing with physical models or LES. SRT assumes the movement of individual fluid parcels near a surface may be represented as a stochastic process driven by a turbulent flow field away from the surface, which is comparable, at least conceptually, to air parcels moving to and from a porous forest canopy exposed to the open atmosphere. SRT has been used to estimate the fluxes of scalar quantities to and from forests (Katul et al., 2013; Paw U et al., 1995). Under certain SRT assumptions, it has been shown that residence times can be well approximated using distributions in the gamma family (Gon Seo and Kook Lee, 1988; Haghighi and Or, 2013, 2015; Katul and Liu, 2017; Zorzetto et al., 2021). We hope a similar approach may be used to estimate $\alpha$ for other forest types, for example, by using LES to calculate $\tau$ across a variety of realistic forests (i.e., including openings, edges, and horizontally heterogeneous structure).

We reiterate here that the above discussion does not include the effect of very strong winds on $\tau$ (see section 2.3), which also lends itself to further testing with LES. We expect $\alpha$ to increase slightly in strong winds, when the reconfiguration of tree crowns allows energetic gusts to penetrate further and more regularly into the forest canopy. The behaviour of $\tau$ across atmospheric stability regimes is more difficult to parametrise. We obtain good fits on both our leaf-on and leaf-off observations in unstable conditions using $\beta = 2h_c/w_*$ in Eq. (7), where $w_* = (g\overline{\overline{w'T_s'}}h_c/\overline{\overline{T}}_s)^{1/3}$, the Deardorff convective velocity scale (defined locally). However, we do not have sufficient spatial resolution in our observations to determine whether this result is robust across a range of unstable conditions, or whether it is just a consequence of the flexibility of the inverse-gamma distribution. In stable conditions, turbulence is dominated by turbulent structures that are intermittent in space and time. These intermittent structures can induce complex flow patterns that do not lend themselves to scaling analysis. The following subsection discusses evidence of this complex behaviour and its implications for residence times of air in the forest canopy.

**3.7 Longer residence times evidenced by evening venting events**

On some evenings during the leaf-on period, we observed 'bumps' in the $[CO_2]$ time series shortly after fumigation was shut down, whereby the $[CO_2]$ decays to a$[CO_2]$, rises again by tens of µmol mol$^{-1}$ for several minutes, before decaying again to a$[CO_2]$. Figure 10 shows a representative example from 17 August 2020. Pools of $CO_2$ can accumulate naturally in forests, e.g., from soil respiration on calm, humid nights, creating anomalously high carbon flux values when the pools are vented from the canopy (Cook et al., 2004). The venting of natural pools typically occurs in the early hours of the morning, after the $CO_2$ has had time to accumulate in the stable nocturnal conditions (Cook et al., 2004), and can last for several hours. Here, the bumps occur shortly after shutdown, last for no more than a few minutes, and occur only in the fumigation arrays. We therefore believe these bump signals are evidence of the venting from the canopy of trapped fumigation $CO_2$ within the canopy, rather than of natural pools (although without isotope analysis it is not possible to conclude with absolute certainty). To investigate these bumps further, we filtered the data according to the following criteria: at least 15 minutes after the shutdown time, the $[CO_2]$ in one or more of the arrays rises by $\geq 15$ µmol mol$^{-1}$ from the a$[CO_2]$ for $\geq 3$ minutes. These criteria are somewhat arbitrary but serve to distinguish the signal from the inevitable noise as the $[CO_2]$ decays to a$[CO_2]$. These criteria identified 41 days with bump events during the leaf-on period, from a total of 452 observation days (i.e., about 9% of the time). Using these criteria, no bumps occurred in the leaf-off period.

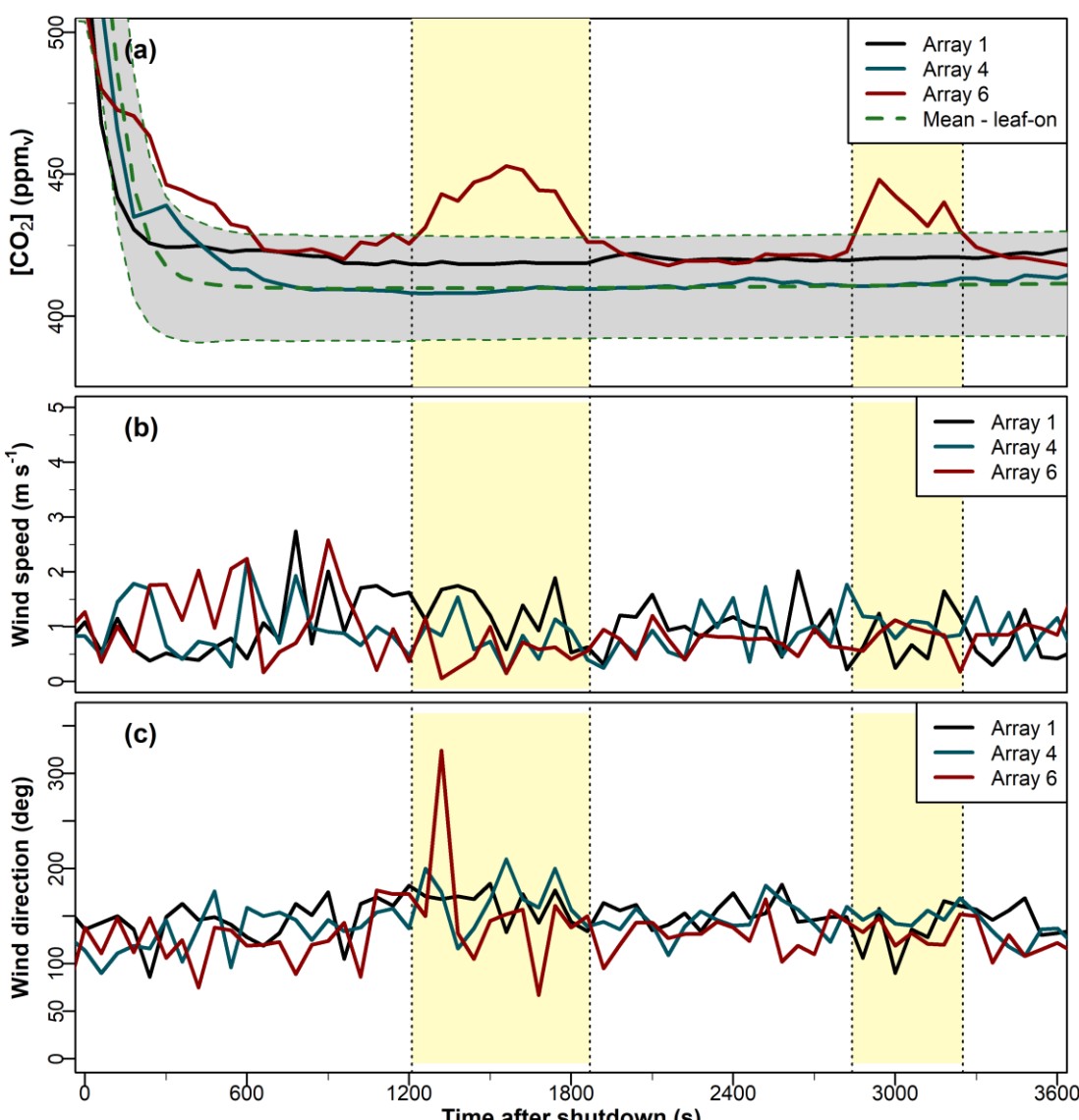

Figure 10: Timeseries of 1-minute averages in (a) the $CO_2$ mixing ration, denoted [$CO_2$] (section 2.1), (b) wind speed, and (c) wind direction after shutdown on 17 Aug 2020. In panel (a), the dashed green line shows the mean [$CO_2$] at each 1-minute time step after shutdown for the leaf-on period (section 2.2 for observational details). The grey shaded confidence interval in (a) shows one standard deviation either side of the mean. The standard deviation is presented here to emphasise that these venting events are not simply symptoms of the variability of the [$CO_2$] observations; on this dataset, it is larger than the other two measures of statistical variability used in this paper (the IQR and $D_{med}$). The shaded yellow rectangles indicate the approximate duration of the venting events.

The bumps occurred only when wind speeds were low, all with $\bar{\bar{u}} < 1.5$ m s$^{-1}$ and typically with $\bar{\bar{u}} < 1$ m s$^{-1}$. This was a necessary but not sufficient condition; there were days with weak winds but no bump events in the [$CO_2$] time series. These bump events may be caused by $CO_2$-rich air being trapped within the dense canopy, particularly when the surrounding atmospheric conditions are very stable, which can occur on evenings with low winds and strong stratification from radiative cooling. The venting occurs when intermittent turbulent structures interact with the forest airspace (whose local stability may differ to that of the surrounding atmosphere). In very stable conditions, boundary-layer turbulence is intermittent in space and time, and may arise from with a variety of phenomena, such as differential heating, top-down turbulent bursts, or larger 'submeso' motions such as microfronts and short gravity waves (Mahrt, 2014; Wharton et al., 2017). These turbulent structures tend to be highly localised, which could explain why the bumps in our time series rarely occurred in more than one array at any one time, even though they typically last for 10 minutes or so (Figure 10). Detecting intermittent turbulent structures around forests requires dense networks of 3D anemometers throughout the canopy, which BIFoR FACE did not have for most of our investigation period (we have recently installed several anemometers within the forest for future investigations). However, on a few occasions, the meteorological towers around the edge of the forest were able to detect the presence of submeso structures (see the case study in Appendix C).

## 4 Conclusions

Our opportunistic investigations of fumigation data from the BIFoR FACE facility provide the first observational evidence of residence times of air in the upper canopy of a deciduous forest. Residence times in the upper half of the forest canopy vary strongly with atmospheric stability, and their statistics differ significantly when the forest is in leaf compared with when it is not. Our dataset shows that air parcels in the BIFoR FACE facility have the following characteristics:

1. When the trees are in leaf, we found median daytime residence times, $\tilde{\tau}$, are around twice as long ($\tilde{\tau} \approx 70$ s) as when the trees are not in leaf ($\tilde{\tau} \approx 34$ s). The spread in the values of $\tau$ is over twice as large when the trees are in leaf versus when they are not in leaf, e.g., median absolute deviation, $D_{med} \approx 51$ s for leaf-on and $D_{med} \approx 20$ s for leaf-off.

2. For chemically reactive tracers, such as BVOCs, released in the upper canopy during daytime, our results suggest the molecules are unlikely to have time to react within the forest unless their chemical reaction time scale $\tau_{chem}$ is in the order of a few minutes or less.

3. Our results agree with Lagrangian modelling studies that the distributions of $\tau$ are strongly positively skewed (e.g., Figure 4). For these types of distributions, average values are not representative of the population as a whole. Where possible, future investigations should report the distributions of residence times, or at least a variability measure to accompany average values. Median values, accompanied by the interquartile range or $D_{med}$, are preferable to the mean and standard deviation because the former are more robust measures of highly positively skewed distributions.

4. The PDFs of residence times can be closely approximated using the inverse-gamma distribution. Models using eddy-diffusivity turbulence closure generate plausible average values but probably overestimate the probability of very long residence times in the upper canopy (i.e., the PDF tails are too heavy). We find the canopy turnover timescale, $\tau_c = h_c/u_*$, provides a good approximation for the scale parameter of the inverse-gamma distribution, with the shape parameter a function of the forest's structure. Although outside the scope of the present study, we suggest that careful testing using physical models or LES will be able to generate robust residence time parametrisations based on simple gamma-like distributions, where the shape and rate/scale parameters can be estimated from variables such as the LAI or wind-velocity statistics, which are available at most forest research sites and, increasingly, at all forest locations from remote sensing.

5. Residence times increase with increasing stability, as does the spread in their values. In unstable conditions, long residence times are much less common than they are in near-neutral or stable conditions. In neutral and unstable conditions, the effect of turbulence levels on the residence times can be approximated $\tau \propto u_*^{-\gamma}$. Our data show $\gamma \approx 1$ in unstable and neutral conditions, but $\gamma \approx 0.8$ in stable conditions.

6. Very long residence times (tens of minutes to hours) can occur in the evening boundary-layer transition when the trees are in leaf. These are evidenced in our data by the venting of trapped $CO_2$ from the canopy long after FACE fumigation has been shut down for the day. This behaviour occurs on a little fewer than 10% of the days with suitable meteorology in our dataset. Cook et al. (2004) report nocturnal venting of pooled $CO_2$ over the course of several hours, which is different from what we see here. We are not aware of any other observational evidence of these brief evening venting events, which typically last around 5–20 minutes and are highly localised, usually in a single fumigation patch. The evening venting events occur only in low winds. We suspect they are evidence of the decoupled forest air space interacting with intermittent turbulent structures in very stable conditions. We found a single case study of a warm microfront, a type of 'submeso' atmospheric motion, causing venting of the forest air space (Appendix C), but the causes of the majority of venting events are not known.

7. The observation of these venting events, and the long residence times they imply, fits with previous field studies that nocturnal residence times are often in the order of several hours, rather than the few minutes typical of daytime values (Martens et al., 2004; Rummel et al., 2002). The stable boundary layer, particularly during the evening and at night, remains poorly understood. Further investigations of nocturnal residence times are needed to understand how physical

processes determine in-canopy chemistry, e.g., the mixing ratios of monoterpenes in boreal forests are at their highest at night, but those for isoprene are at their lowest (Hakola et al., 2012). These investigations need to be centred around robust observations and physical experiments—nocturnal exchange is dominated by intermittent turbulence that is difficult to constrain in numerical models (Bannister et al., 2022; Mahrt, 2014; Sterk et al., 2016).

*Data availability.* The time-averaged measurements, supporting README files, and visualisation code for this work are deposited in the following open-access repository: https://doi.org/10.25500/edata.bham.00000836. The raw BIFoR FACE measurements are publicly accessible at https://doi.org/10.25500/edata.bham.00000564 – see Hart et al. (2020) and MacKenzie et al. (2021) for further detail.

*Author contributions.* EJB conceived of the study, wrote the processing code, conducted the formal analysis and visualisation, and wrote the original draft, under the supervision of ARMK, MJ, and XMC, who provided regular feedback on the methodology and original draft. KMH, MJH, GC are responsible for the operation of the BIFoR FACE facility and the curation of the data arising from it. All authors contributed to reviewing and editing the manuscript.

*Competing interests.* The authors declare that they have no conflict of interest.

*Acknowledgments.* The BIFoR FACE facility is a research infrastructure project supported by the JABBS Foundation and the University of Birmingham. The authors gratefully acknowledge additional funding from the John Horseman Trust, the Ecological Continuity Trust, and the UK Natural Environment Research Council (NERC, grant NE/S015833/1) in support of this work and the wider research at the BIFoR FACE facility. It is EJB's pleasure to acknowledge the Natural Environment Research Council (NERC) for funding through a CENTA studentship (grant NE/L002493/1). Photograph of the infrastructure in Figure 1 (bottom right of figure) taken by Andrew Priest Photography. The authors thank Chantal Jackson for her help in drawing and compiling Figure 1. We conducted the data analysis and statistical calculations using R (R Core Team, 2021), including the openair package (Carslaw and Ropkins, 2012), and Python. We are grateful to Tobias Gerken, and an anonymous reviewer, whose thoughtful comments helped us refine and present the ideas in this manuscript.

## Appendix A – Estimating $K_{eq}$ in Eq. (1)

A variety of methods can be used to estimate $K_{eq}$ around vegetation (Haverd et al., 2009; Monteith and Unsworth, 2008). We use GCF17's simple parametrization

$$K_{eq} = T_l g(LAI) u_* h_c, \tag{A1}$$

where $h_c$ is the mean height of the canopy, $u_*$ is the friction velocity measured at a height $h_c$, $T_l$ is the Lagrangian integral time scale normalised by $h_c/u_*$, and $g(LAI)$ is a function that adapts the profile of the vertical velocity variance to the canopy structure such that

$$g(LAI) = c_1^2 \frac{2c_2 - 4\exp(c_2) + \exp(2c_2) + 3}{2c_2(\exp(c_2) - 1)^2} \tag{A2}$$

where $c_1$ and $c_2$ are modelling constants, with $c_1 = 0.9$ and $c_2 \approx -0.5$–$1.5$. GCF17 use $T_l = 1/3$ from Raupach (1989), but we obtain better results on our data using the estimate of Haverd et al. (2009) such that

$$T_l = c_4 \frac{1 - \exp\left(-c_3 z_{rel}/h_c\right)}{1 - \exp\left(-c_3\right)}, \tag{A3}$$

where $c_3 = 4.86 \pm 1.52$ and $c_4 = 0.66 \pm 0.1$, which gives $T_l \approx 0.6$. Taken together, these assumptions obtained $K_{eq} = 1.2$ m$^2$ s$^{-1}$, which was used in Eq. (1) to generate the GCF17 PDF in Figure 3, for example.

**Appendix B**

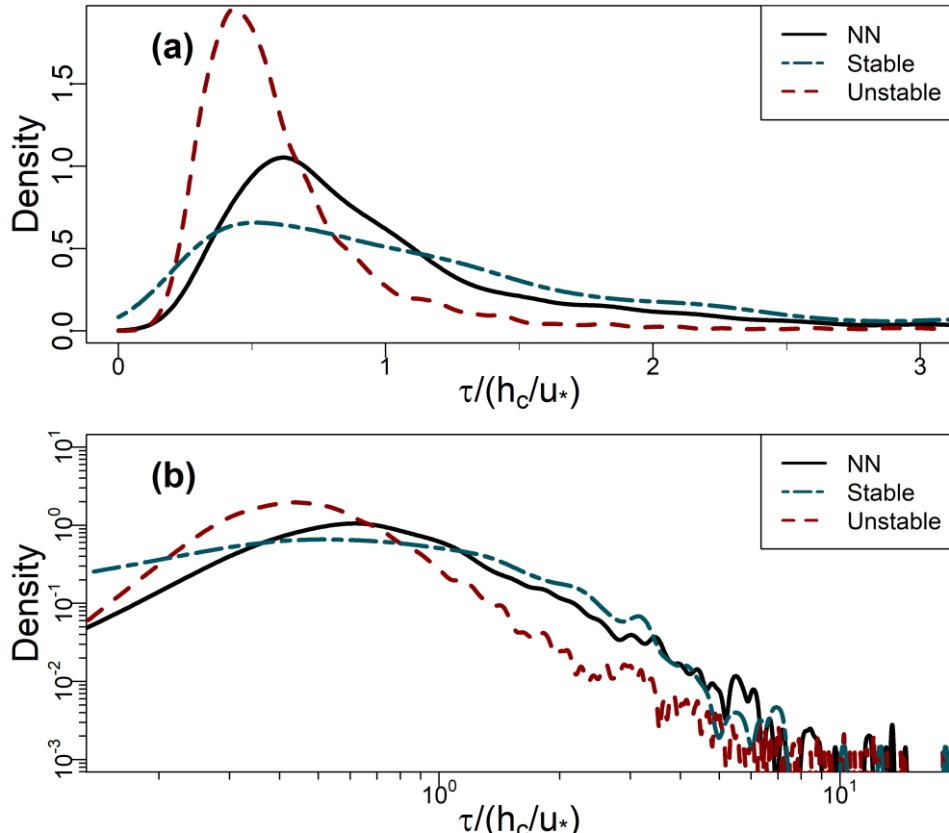

**Figure A1: (a) and (b) PDFs of residence times (section 2.3) binned by stability class for the leaf-off period (section 2.2), plotted on linear and logarithmic axes (base 10), respectively. NN denotes near-neutral conditions. See section 2.6 for definitions and Figure 7b, c for analogous results from the leaf-on period.**

**Appendix C – Case study of venting from submeso motions**

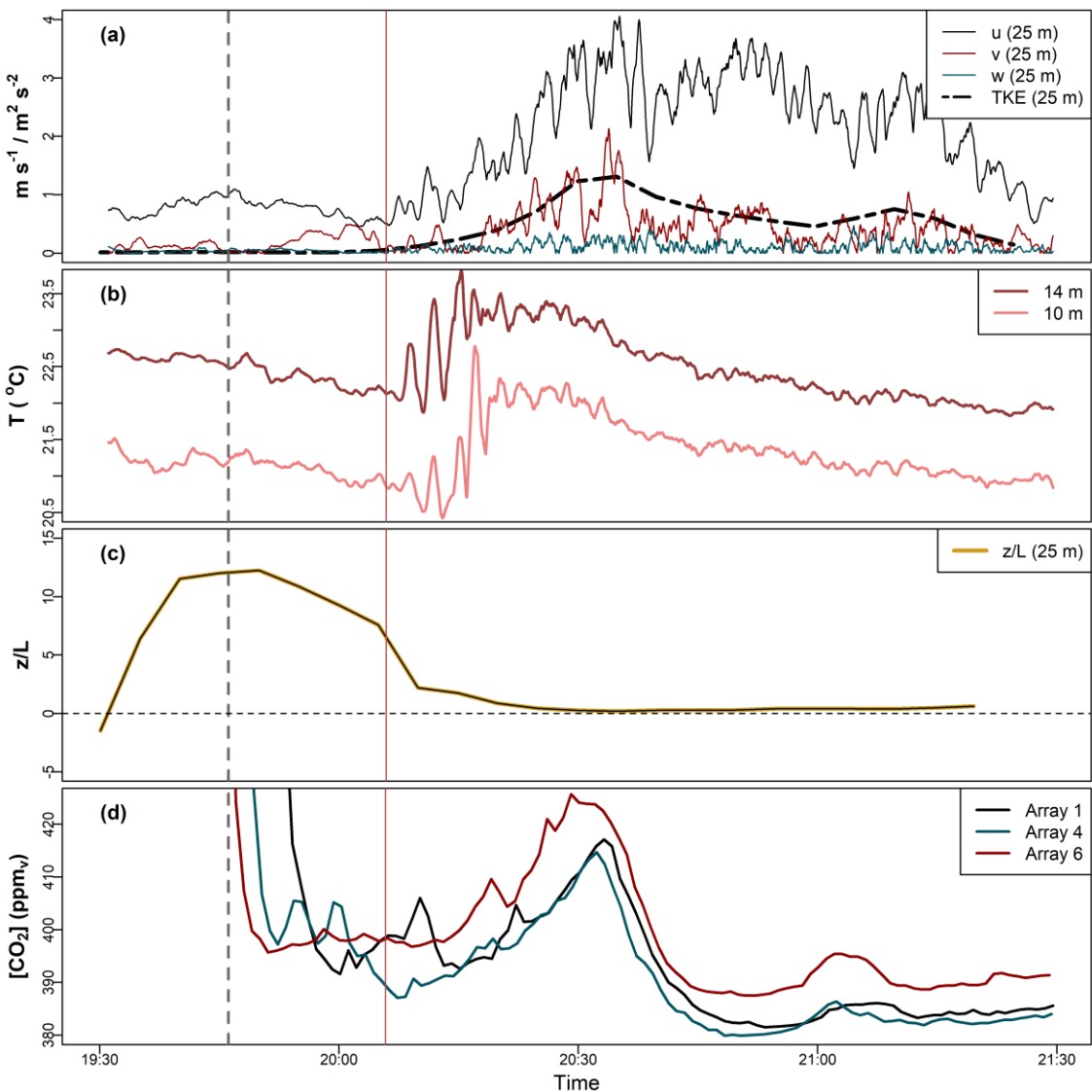

**Figure A2: Timeseries after shutdown on 27 Aug 2019 of (a) the magnitude of the velocity vector components (m s-1) and TKE (m² s⁻²) at *z* = 25 m, (b) T (°C), (c) *z/L* (where *L* is the Obukhov length, section 2.5), and (d) [CO2]. The dashed grey vertical line indicates the shutdown time and the solid red line the approximate arrival time of the warm microfront described in the paragraph below. The high-resolution measurements in (a)–(c) are from on Met 1, at the southern edge of the BIFoR FACE facility (Figure 1 and section 2.1). (a) and (b) show 1-min rolling means.**

Figure A2 presents a case study from 27 August 2019, which was a warm, cloudless day with weak southerly winds. Around sunset, and therefore fumigation shutdown (marked with the dashed grey vertical line in Figure A2), the wind speed was very low—less than 1 m s⁻¹ at $z = 25$ m at the forest edge (Figure A2a) and almost zero within the canopy. The TKE was close to zero (Figure A2a). A strong temperature inversion formed (Figure A2b), with the temperature at $z = 25$ m a further few degrees warmer than at $z = 14$ m (not shown). The air around the forest was very stable, with $z/L \approx 10$–15 (Figure A2c). Around 20 minutes after shutdown, a warm microfront reached the southern edge of the forest, marked with the solid vertical red line in Figure A2 and visible in a sharp increase in temperature near the ground (Figure A2b). The passage of warm microfront leads to increased local wind speed and turbulent intensity, and decreased atmospheric stratification (Mahrt, 2019). These changes can be seen in Figure A2a, where the horizontal wind speed and the TKE increase quickly, and Figure A2c, which shows the stability decaying quickly from very stable to approximately neutral (i.e., $z/L \approx 0$). The increased wind speed and turbulence cause trapped $CO_2$ to be vented from the forest in all the fumigation arrays (Figure A2d). As well as providing interesting micrometeorological case studies, these venting events provide observational evidence that residence

times can be much longer in stable evening conditions compared with the average daytime values, e.g., at least 20–30 minutes in the example in Figure A2 and nearly 60 minutes in the example in Figure 10.

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
