# Peer review of "Residence times of air in a mature forest: observational evidence from a free-air CO2 enrichment experiment"

_Atmospheric Chemistry and Physics, 2022_

## Author Response (AR3)

Bannister, E. J., Jesson, M., Harper, N. J., Hart, K. M., Curioni, G., Cai, X., and MacKenzie, A. R.: Residence times of air in a mature forest: observational evidence from a free-air $CO_2$ enrichment experiment, *Atmos. Chem. Phys. (2023)* https://doi.org/10.5194/acp-2022-318.

This document contains the combined author responses to comments on acp-2022-318. This work is distributed under the Creative Commons Attribution 4.0 License.

**Authors' responses to reviewers' comments on acp-2022-318 (reviewers: Tobias Gerken and one anonymous referee)**

**The authors thank Tobias Gerken (RC1) and the anonymous reviewer (RC2) for their thoughtful comments on our manuscript. We found the comments valuable, and they helped us refine and present the ideas in our manuscript.**

**We include detailed responses to RC1's comments below. These are preceded with '>>>' in case the bold formatting is stripped in the online version.**

**As a summary of the main changes to the manuscript, we have:**

- **Clarified our definition of a 'residence time' of air in a forest canopy. We now use 'air-parcel residence times' only in the context of Lagrangian investigations, and 'residence time of air' in the context of our results (which are derived using a Eulerian mass-balance approach). This change allows a more balanced comparison of our**
**results and those in earlier papers (most notably, to GCF17).**

- **Included more discussion of the limitations in our method, such as the possibility of non-negligible advection from the control volumes within which the residence times are calculated.**

- **Included an illustrative example, using the reaction timescale of isoprene, of the influence of short residence times on Damköhler numbers (and how they may respond to changes in atmospheric stability).**

- **Improving several of the figures, and making the figure captions stand alone, so that the key information can be found by readers skim-reading the paper.**

- **Including more detail on the complexity of density and morphology in real forest canopies. The effect of canopy structure on residence times has not been tested in the field, but we include references to investigations in idealised numerical models (lines 83–95).**

- **Changed references to statistical 'dispersion', which could potentially be confusing in this context, to 'spread' or 'variability'.**

- **Included a short introduction to the strengths and weaknesses of the three methods of statistical variability used in this manuscript.**

**Specific comments**:

Title: In the light of my general comment 1, it might be better to remove the 'parcel' from the title.

**>>> This a fair comment and we have adjusted the title of the manuscript**.

L 67: "Gerken et al. (2017) (hereafter GCF17) offer ..." > I suggest to also reference Katul et al. (2005) in this context, since their model was used as a starting point and includes similar assumptions. One should also note that this formulation was only proposed for neutral conditions.

**>>> We have included this reference at lines 64–65, thank you.**

L 87: "In an LES investigation of flow over forested hills, residence times of Lagrangian air parcels emitted in the lower part of the canopy were shorter than those moving over flat terrain (Chen et al., 2019)." > I suggest to expand on this since the impacts of terrain are very important for real world applications and there is ample evidence (albeit anecdotal) for preferential venting due to terrain.

**>>> We have expanded this section, including references to relevant observational and modelling studies (lines 97–106). However, a detailed discussion of the various flow phenomena that occur in hilly terrain is out of scope of this study (see the included references for further detail).**

L90: "Researchers have also used Eulerian frameworks to investigate residence times in forests." > It might be good in this context to discuss some of the limitations on Eulerian vs. Lagrangian methods regarding their implications for air chemistry. This also goes along with my general comment on the comparability of Lagrangian and Eulerian approaches. Given that our parcel residence time is a fundamentally Lagrangian process, the Eulerian description has some limitations such as that it is in my mind a mean air-residence time, which can underestimate the tail ends, which might be important for air chemistry.

**>>> This is an important point. We have included some discussion of the strengths and weaknesses of the two approaches at 111–124. We have also adjusted the wording of the manuscript to refer to 'air-parcel residence times' only in the context of Lagrangian investigations, and 'residence time of air' in the context of our results (which are derived using a Eulerian mass-balance approach).**

L200: "because broadleaf forests uptake little carbon below this threshold" > this sounds off. 'because carbon uptake is negligible' (?)

**>>> We have clarified this statement, as suggested, at lines 232–233.**

Section 2.3: What is the time scale of the \tau calculation? (P.S. I see that this is answered in the data processing section. I suggest to move this forward). Additionally, and given the importance of release height pointed out in GCF17, it would also be good to hear more about at what heights $CO_2$ is being releases.

**>>> We have moved the timescale of the residence calculation to section 2.3, as suggested (lines 281–288). We have also provided a little more background on the heights of the fumigation at BIFoR FACE in sections 2.3 and 2.4 (lines 277–279 and 321–325).**

L 237-240" "Therefore, rather than trying to assign a numerical value to $F_{out(hor)}$, we identify meteorological conditions under
which $F_{out(top)} \gg F_{out(hor)}$, and therefore $\tau = M_{CO_2}/F_{in} \approx M_{CO_2}/F_{out(top)}$. Figure 3 presents probability density functions of $\tau$ during the lowest 50% of wind speeds of the leaf-on period (solid black), during the highest 25% of wind speeds of the leaf-on period (dashed),... " > This seems like an abrupt transition. It might be good to give provide a sentence or two on how these are related to advection. On a broader note, I appreciate the advection problem in the sense that this is something that has been challenging in high vegetation with $CO_2$ accumulation within the canopy airspace.

**>>> The mean wind speeds at z = 25 m are no more than around 2 m/s for the lower 50% of our observations, on which we derived the residence time. Because the wind speed decays exponentially with height within forest canopies, the speed at the height of the release (around 15 m, see comment below) is likely to be very low. On this basis, advection will of course be non-zero, but is likely to be small (see lines 264–276). Unfortunately, we do not have a direct measure**

**of advective fluxes at the BIFoR FACE facility, and such measurements have proved to be very difficult even with dense networks of sensors (Aubinet et al., 2010). We therefore cannot make this point unequivocally. However, we are further comforted by long-term analysis of the BIFoR FACE observations, which suggest that contamination events between the arrays are rare— i.e., passage of CO2-rich air from the fumigation arrays to the control arrays (Hart et al., 2020).**

L 241" zrel > I am wondering whether it would make sense to adjust z_rel, given that it is not clear to me what the real release height of the $CO_2$ is would be to minimize the difference between the observational results and the theoretical result by adjusting z_rel.

**>>> We tested a range of values for $z_{rel}$, and obtained the best fit between our data and GCF17 using $z_{rel} \approx 15$ m. However, see our comments as to the fumigation heights at BIFoR FACE (lines 277–279 and 321–325)—a single value for $z_{rel}$ is more difficult to define for our observations than for the Lagrangian parcels in GCF17's model.**

Figure 3a: It seems to me that all the curves in Figure 3 should have the same integral. Could you confirm this and check
whether all curves are properly normalized, since it seems (by eyeballing) that the GCF17 might not have the same area under the curve.

**>>> Thank you. We have checked, and confirm the normalisation of the PDFs in Figure 3 is correct (as for the PDFs elsewhere in the paper). We agree the left tail of the GCF17 in Figure 3 looks slightly odd, particularly on the log axes.**
**This is caused by the smoothing algorithm in the kernel density estimation for very small values of $\tau$, but does not affect the conclusions in this paper.**

Figure 6 should have a colorbar and possibly a trendline to better gauge the underlying density distribution.

**>>> We have updated Figure 6, using filled contour lines rather than hexagons to display the 2D density plots. We have included a colour scale, as you correctly suggest.**

The stability classes in Section 3.3.2 should probably be moved to methods section.

**>>> Thank you for this suggestion. We have moved the definitions of the stability classes to the new section 2.6, and have updated the cross references.**

L 384: "The distributions of $\tau$ remain positively skewed for each stability class (e.g., the right whiskers are longer than the left in Figure 7a)." > It might be a good idea, here and in general to report the skewness.

**>>> Good idea. We now report the skewness of the distributions in Table 2.**

Section 3.4.: I am not sure how informative this section and the associated figure is. I think that it is important to discuss the edge effect and impacts of heterogeneity, but I am not sure whether this section currently does this in the optimal way. Especially since I think that Figure 8 is pretty hard to read. What dominates the differences in the different wind sectors. Is it heterogeneity or some other influence such as time of day coupled with stability?

**>>> We have clarified the caption to Figure 8, to make the plot easier to interpret. Edge effects are important for both the flow and for forest ecology. However, their effects on residence times are difficult to generalise because they are very site specific, save for very general comments, such as regions of high turbulence (e.g., upwind edges or clearings) having shorter residence times. Our findings in section 3.4 did not materially vary with time of day or stability—see (bearing in mind that we are considering only daytime residence times).**

L445: "However, although GCF17's model generates modal values similar to those we observed, it appears to overpredict the likelihood of long residence times in the upper canopy." > this might be true, but also there might be the issue of comparing an essentially Eulerian and Lagrangian method (see general comments).

**>>> This is a fair comment. We have qualified our findings (see 517–541), including noting that the comparison is not completely like-for-like (i.e., Lagrangian air parcels in GCF17 and, in our study, a mean (five-min) residence time of air in a control volume)**

L463: "These eddies create significant turbulent transport, meaning that the eddy-diffusivity model underestimates turbulent forest-atmosphere exchange in the upper canopy and therefore overestimates residence times." > Turbulent diffusivity approaches do have issues within forest canopies. One thing to note about GCF17 is the fact eddy diffusivities are estimated from the LES and adjusted for the release height by taken the mean modeled diffusivity (either arithmetic or geometric mean) between release height and canopy top. It is not clear to me that this would lead to an effective under estimation of turbulent transport. Some additional thoughts by the authors would be appreciated.

**>>> We agree that the use of LES (as in GCF17) navigates some of the issues with earlier Lagrangian stochastic models, such as the need to specify a priori profiles of turbulence statistics, which are usually not realistic. LES remains the best method for turbulence-resolving models of exchange in forest canopies—at least for studies on the scale of metres to km, and seconds to hours—and GCF17 is an example of the technique being used well.**

**However, LES is by no means perfect—e.g., it requires a sub-grid parametrisation scheme, which are seldom tuned to flows in vegetation, and may under-simulate certain aspects of the flow in forests, such as the penetration of sweeps and gusts, or the boundary-layer TKE above the canopy. Further, most LES models of forests (including that in GCF17) do not resolve the inherent patchiness found in reality (gaps, clearings, roads, different-aged trees etc.), let alone landscape fragmentation. These features would likely reduce the probability of long residence times of air in forests, relative to homogeneous models, because turbulent mixing would be stronger in reality than in the models. However, to our knowledge, this has not been tested directly. See lines 533–541.**

Section 3.7: This section seems a bit tacked on in the sense that it is not clear how it relates to the previous section of the model, especially given that the main conclusion from the previous section seemed to point to an overestimation of residence times in GCF17. While the information presented here is an interesting case study, it might make sense to either tie this directly to the analysis before or to remove/ move to an appendix.

**>>> We agree that the manuscript is more reader-friendly by focussing on the residence-time analysis. However, we consider that the case study that was previously in section 3.7—an unusual example of a microfront causing venting of the canopy airspace—could be valuable to the community. We have therefore retained the case study, but have moved it to a new Appendix C (renumbering Figure 11 to Figure A2, and updating the supporting text).**

On a side note, and also with respect to long residence times: the original GFC17 study was motivated by air exchange within Amazon rainforest canopies with large LAI and limited penetration of turbulent eddies into the lower half of the canopy. Evidence for the limited coupling of canopy airspace to the above canopy air can be for example found in Freire et al. (2017).

**>>> Noted with thanks.**

**Authors' responses to referee's report #1 following the initial round of revisions to our manuscript.**

The authors thank Tobias Gerken (RC1), the anonymous reviewer (RC2), and the Christoph Gerbig, the handling editor, for their continued effort and helpful suggestions.

In accordance with RC1's suggestions, we have:

1. Moved the illustrative example of the Damköhler number to the main results/discussion section (section 3.2). This avoids introducing new material in the conclusions.
2. Improved the discussion of the effects of canopy density on the residence time of air (lines 83–93).

We also made a small number of stylistic changes (correcting 'compared to' to 'compared with' and using hyphens where compound nouns serve as adjectives).